# Just Select Twice: Leveraging Low Quality Data to Improve Data Selection

## Abstract

Data valuation is crucial for assessing the impact and quality of individual data points, enabling the ranking of data by importance for efficient data collection, storage, and training. Many data valuation methods are sensitive to outliers and require a large level of noise to effectively distinguish low-quality data from high-quality data, making them particularly useful for data removal tasks. In particular, optimal transport-based methods exhibit notable performance in outlier detection but show only moderate effectiveness in high-quality data selection, due to their sensitivity to outliers and insensitivity to small variations. To mitigate the issue of insensitivity to high-quality data and facilitate effective data selection, in this paper, we propose a straightforward two-stage approach, JST, that initially does data valuation as usual, but then performs a second-round data selection where the identified low-quality data points are designated as the validation set to perform data valuation again. In this way, high-quality data become outliers with respect to the new validation set and can be naturally identified[1]. We empirically evaluate an instantiation of our framework based on optimal transport method for data selection and data pruning on several standard datasets and our framework demonstrates superior performance compared to pure data valuation, especially under small noise conditions. Additionally, we show the general applicability of our framework to influence function based and reinforcement learning based data valuation methods.

## 1 Introduction

Access to large, high-quality datasets is essential in machine learning. However, in real-world data collection and curation pipelines, individual data points often inherit different levels of quality and vary in their importance on the impact of training (Liang & Zou, 2022; Sorscher et al., 2022; Liu et al., 2021). Therefore, it is critical to understand and assess such properties of data and to effectively prioritize highly valuable data sources for subset selection. This can assist practitioners in improving model performance efficiently (Karr et al., 2006; Jiang et al., 2023) and make strategic, cost-effective decisions in data marketplaces and exchanges (Olaleye & Adusei, 2024).

There have been considerable recent efforts to develop various data valuation methods aimed at evaluating individual data points and assigning a value to each one (Jiang et al., 2023; Sim et al., 2022). This can help to quantify differences between points and rank them based on their assigned value, establishing a certain order of quality or importance in the training process. For instance, methods based on optimal transport (Just et al., 2023) and influence functions (Koh & Liang, 2017) employ sensitivity analysis to quantify the impact of individual data points on dataset distance and training outcome, respectively. Additionally, importance weights can be obtained through reinforcement learning (Yoon et al., 2020) to evaluate individual data points.

In essence, these data valuation methods should naturally provide a metric for high-quality data subset selection. However, unfortunately, most data valuation methods have predominantly excelled in scenarios with large noise, including intensely corrupted samples (Just et al., 2023), randomly flipping labels (Just et al., 2023; Koh & Liang, 2017; Yoon et al., 2020), or domain adaption due to

---

[1]To be clear, in typical data valuation methods, outliers are considered with respect to a clean validation set, with noisy data treated as outliers. In our framework, the validation set is noisy, so high-quality clean data are considered outliers.

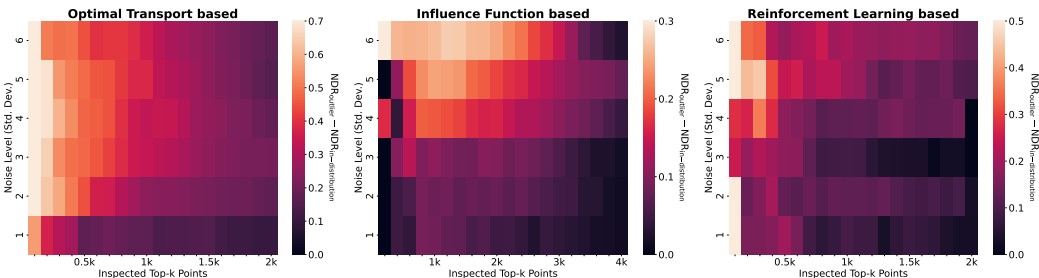

Figure 1: Sensitivity comparison between outliers and in-distribution data. We inject white noise into 50% of the CIFAR-10 training set with different noise levels and inspect top k data points for each data valuation method from lowest values for outliers and highest values for in-distribution data, respectively. We evaluate the difference between the normalized outlier detection rate ($\mathrm{NDR_{outlier}}$) and the normalized in-distribution data detection rate ($\mathrm{NDR_{in\text{-}distribution}}$). The outlier data detection rate is much larger than in-distribution data detection rate, reflecting higher sensitivity of these data valuation methods to outliers.

substantial mismatch between training and target domains (Koh & Liang, 2017; Yoon et al., 2020). In particular, the optimal transport-based method in (Just et al., 2023) demonstrates strong performance in outlier detection but exhibits only moderate effectiveness in selecting high-quality data due to its well-known property of sensitivity to outliers and insensitivity to small variations (Villani, 2021).

As demonstrated in Figure 1, the sensitivity of these data valuation methods to outliers is much larger than the high-quality data. Therefore, in this paper, we address the following question: *How can we improve the sensitivity of data valuation methods to high-quality data and make them better when applied to data selection?*

To this end, we propose JST (Just Select Twice), a data subset selection framework that augments existing data valuation methods by leveraging outliers detected by these methods, as illustrated in Figure 2. Our key insight is that incorporating a second-round subset selection, using detected outliers as the new validation set, allows high-quality data to be identified as outliers with respect to the new validation set in this new context. This approach enhances the sensitivity of data valuation methods to high-quality data, establishing a more meaningful order of "importance" or "quality" for subset selection. Even with the inclusion of numerous high-quality data points within the new validation set alongside low-quality data in the second-round subset selection due to the non state-of-the-art performance of data valuation methods, we observe that the signals from such a validation set continue to suffice in achieving superior performance in subset selection compared to pure data valuation methods.

Before diving into the details, we summarize our contributions as follows: (1) JST, a novel and straightforward two-stage framework augmenting existing data valuation methods, rendering them more applicable for data subset selection tasks. (2) We empirically evaluate an instantiation of our framework based on optimal transport method for data selection and data pruning on six standard datasets, showing its superior outperformance to pure data valuation, particularly in small noise settings. We also demonstrate its general applicability to influence function and reinforcement learning based data valuation methods.

## 2 BACKGROUND AND RELATED WORK

We provide a small amount of background and describe related work.

**Data Selection.** Active learning and coreset construction are two widely used methods for data subset selection. These methods aim to identify the most representative training data points. Active learning involves iteratively choosing points to label from a large unlabeled dataset based on the model's uncertainty or other heuristics, such as the entropy of predicted class probabilities (Sener & Savarese, 2018; Coleman et al., 2020). Recently proposed data selection methods in continual learning are related to these approaches, determining examples to be stored or labeled for ongoing model training (Castro et al., 2018; Aljundi et al., 2019).

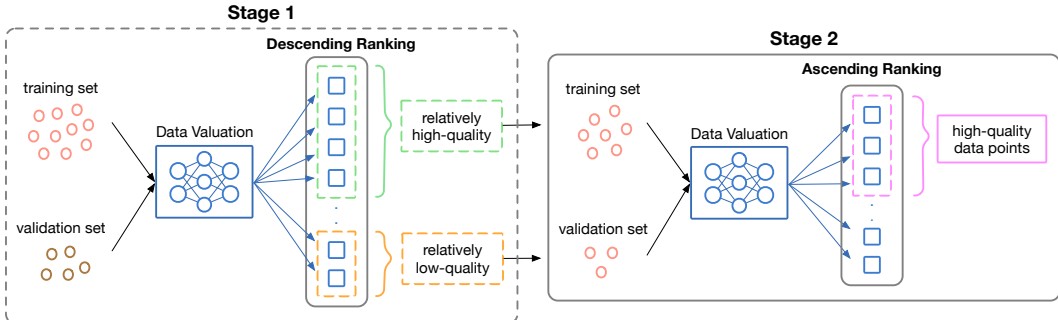

Figure 2: Illustration of JST framework. In stage 1, we perform data valuation as usual and select relatively low-quality data as the validation set. In stage 2, we perform data valuation on the remaining training samples and the new validation set. In this way, high-quality data stands out as outliers with low value scores.

In contrast, coreset construction begins with the entire dataset and tries to select a small subset that encapsulates the essence of the full dataset. The goal is to have the model trained on the subset perform approximately as well as the one trained on the entire dataset (Feldman, 2020; Huang et al., 2021). Many works have tackled this setting to accelerate clustering and learning, proposing coresets in k-means and k-medians clustering (Har-Peled & Kushal, 2005), SVMs (Tsang et al., 2005), Bayesian logistic regression (Huggins et al., 2016), and Bayesian inference (Campbell & Broderick, 2019). Recently, newly proposed coreset selection methods have demonstrated efficiency in neural network training (Mirzasoleiman et al., 2020; Killamsetty et al., 2021b;a; Tukan et al., 2023). In addition, related coreset techniques have been applied to active learning (Sener & Savarese, 2018) and continual learning (Tiwari et al., 2022).

**Data Valuation.** Data valuation assesses the contribution of each individual data point to the overall performance of a model, aiming to distribute the validation performance across the training data points (Jiang et al., 2023). Formally, given a training dataset $D_{tr} = \{z_i\}_{i=1}^N$, a validation dataset $D_v$, and a model performance metric *PERF* evaluated on the validation set $D_v$, data valuation methods assign a scalar score to each training sample $z_i$ in $D_{tr}$ to split *PERF* evaluated on $D_v$. For instance, if we define a utility function $U$ over all subsets $S \subseteq D_{tr}$ of the training data as $U(S) := PERF(\mathcal{A}(S))$ evaluated on the validation set $D_v$ with a learning algorithm $\mathcal{A}$, a straightforward method to evaluate the contribution of a training sample $z_i$ is to calculate the leave-one-out (LOO) value, $U(D_{tr}) - U(D_{tr} \setminus \{z_i\})$. This represents the change in model performance when the point is excluded from the training set (Just et al., 2023).

In practice, feasible alternatives to LOO can be used to estimate the impact of the weight change of a data point on the model performance. Two such approaches are optimal transport-based methods (LAVA) (Just et al., 2023) and influence function-based methods (Koh & Liang, 2017). The latter technique employs the Wasserstein distance between training and validation sets as the performance metric, while the influence function-based approach uses the validation loss. Both methods measure the gradient as the scalar value score with respect to the weight change of a data point. In addition, reinforcement learning based-methods (DVRL) (Yoon et al., 2020) learn a weight function to minimize the weighted empirical risk using policy gradients, and so obtain optimal importance weights as the scalar value scores.

Indeed, data valuation is intimately related to data selection methods, but it formalizes the data selection problem by aiming to select a subset of the training set that matches a desired target distribution, as represented by the validation set (Just et al., 2023; Yoon et al., 2020), because data valuation methods incorporate validation performance into individual training samples. This approach is essentially different from active learning and coreset construction, which instead seek to compress the full training dataset. Therefore, data valuation can be applied to data selection in the scenario of robust learning with a mismatch between training set and target set (Yoon et al., 2020). However, as demonstrated in Figure 1, data valuation methods are typically more sensitive to outliers rather than high-quality data. This property makes data valuation an ineffective approach when seeking to distinguish high-quality data at a finer-grained scale, especially when there is a small mismatch between the training and target sets. To address this issue, motivated by the property, we propose

a straightforward two-stage framework, called JST, to make data valuation more suitable for data selection. To the best of our knowledge, our work is the first to leverage this property to augment existing data valuation methods.

## 3 PRELIMINARIES

We briefly detail the components, setup and, notation used to describe our method.

**Data Selection:** We are given a training set $D_{tr} = \{z_1, \ldots, z_n\}$ containing $n$ data points, where each $z_i = (x_i, y_i)$ is drawn from a source distribution $p_{src}(z)$, as well as a validation set $D_v = \{z'_1, \ldots, z'_m\}$ with $m$ data points, where each $z'_i = (x'_i, y'_i)$ is drawn from a target distribution $p_{trg}(z')$ (typically $n > m$). Both sets share the same input-output space $\mathcal{X} \times \mathcal{Y}$ but differ in their underlying distributions, i.e., $p_{src}(z) \neq p_{trg}(z')$.

Given a selection budget $k$ ($k \leq n$), the goal of a data selection procedure is to identify a subset $\hat{D} = \{\hat{z}_1, \ldots, \hat{z}_k\}$ where $\hat{D} \subseteq D_{tr}$, such that the distribution of $\hat{D}$ closely matches the distribution of the validation set $D_v$, to minimize the impact on the learned model. Therefore, the selected subset should approximate the distribution of the validation set, i.e., $P(\hat{D}) \approx p_{trg}(z')$, where $P(\hat{D})$ is the distribution constructed from $\hat{D}$.

**Data Valuation:** Similarly, we are given a training set $D_{tr} = \{z_1, \ldots, z_n\}$ containing $n$ data points $z_i = (x_i, y_i)$ and a validation set $D_v = \{z'_1, \ldots, z'_m\}$ with $m$ data points $z'_i = (x'_i, y'_i)$. As before both sets share the same input-output space $\mathcal{X} \times \mathcal{Y}$.

The goal of data valuation is to understand and distribute the validation performance across training data points. To achieve this, we use a function $\mathcal{V}$ computed over the training set $D_{tr}$ to find a score vector $\bar{s} \in \mathbb{R}^n$ that represents the allocation to each data point, described as follows:

$$\bar{s} := \mathcal{V}(D_{tr}, D_v), \text{ where } \bar{s} \in \mathbb{R}^n. \tag{1}$$

Given a score vector $\bar{s} = [s_1, \ldots, s_n]$ representing scores $s_i$ for each data point $z_i = (x_i, y_i)$ in training set $D_{tr}$, we can express the process of ranking (in descending order) and indexing the $k$ points starting from index $r$ as follows:

$$D_{sel} := \mathcal{R}(\bar{s})[r : r + k], \text{ where } |D_{sel}| = k \text{ and } D_{sel} \subseteq D_{tr}. \tag{2}$$

## 4 JST: JUST SELECT TWICE

We now present JST, a straightforward two-stage approach to augment existing data valuation methods. In the first stage of the process, we perform data valuation as usual and select data points with lowest value scores as the validation set. Then, in the second stage of the process, we perform data valuation on remaining training data points and select the data points of low value scores as high-quality data.

**Stage 1 (Data Valuation).**
Given a training set $D_{tr}$ and a validation set $D_v$, let $D_{sel}$ of size $|D_v|$ represent the selected data points of lowest value scores obtained by the ranking function $\mathcal{R}(\cdot)[|D_{tr}| - |D_v| : |D_{tr}|]$ of data valuation $\mathcal{V}(D_{tr}, D_v)$.

**Stage 2 (Data Selection).**
Substitute the original validation set $D_v$ with $D_{sel}$ obtained in the first step and remove data points in $D_{sel}$ from the training set $D_{tr}$, denoted as $D'_{tr} = D_{tr} \setminus D_{sel}$. Then perform data valuation $\mathcal{V}(\cdot, \cdot)$ again but instead on the changed training set $D'_{tr}$ and validation set $D_{sel}$ and minus the score vector, i.e., $\bar{s'} = -\mathcal{V}(D'_{tr}, D_{sel})$. Finally, the top $k$ high-quality data can be selected by $\mathcal{R}(\bar{s'})[1 : 1 + k]$. We summarize our framework in Algorithm 1.

---

**Algorithm 1** JST Selection

**Input:** training set $D_{tr} = \{(x_i, y_i)\}_{i=1}^n$, validation set $D_v = \{(x'_i, y'_i)\}_{i=1}^m$, data valuation $\mathcal{V}(\cdot, \cdot)$

**Output:** data value score vector $\bar{s'}$

**Stage one:**
Perform data valuation:
    $\bar{s} \leftarrow \mathcal{V}(D_{tr}, D_v)$
Select backward data points:
    $D_{sel} \leftarrow \mathcal{R}(\bar{s})[|D_{tr}| - |D_v| : |D_{tr}|]$

**Stage two:**
Remove training data points:
    $D'_{tr} \leftarrow D_{tr} \setminus D_{sel}$
Perform data valuation again and minus the score vector:
    $\bar{s'} \leftarrow -\mathcal{V}(D'_{tr}, D_{sel})$

---

**Practical Implementation.** In practice, similar to other data valuation and selection methods (Sorscher et al., 2022; Jiang et al., 2023; Just et al., 2023; Xia et al., 2023), we begin by using a neural network trained on the validation set $D_v$ to extract features into a low-dimensional space. This allows us to leverage feature relationships effectively and compute data value scores efficiently. Given that the manually cleaned validation set $D_v$ is often small, a ResNet-18 model (He et al., 2016) is an appropriate choice for feature extraction. However, in the second round, prior knowledge from the clean validation set $D_v$ can erroneously align the new validation set $D_{sel}$ containing low-quality data with the remaining training samples $D'_{tr}$, negatively impacting performance. Given our limited access to noisy data, we find that leveraging a pretrained ResNet-18 model on ImageNet1K (Deng et al., 2009) suffices.

To maintain a perfect match between our algorithm and pure data valuation methods, we use the same number $|D_v|$ of backward training data points $D_{sel}$ in the second round as the validation set, which is always mixed with a non-obvious proportion of high-quality data. However, we find that behavior does not affect the performance of our algorithm. This is because even a small portion of signals from low-quality data is sufficient to pop out high-quality data with low value scores. Therefore, in practice, we can reduce the size of the validation set $|D_{sel}|$ to involve more training samples for selection in the second round without degrading performance.

Ablation experiments aimed at these two notions are deferred to Appendix B.

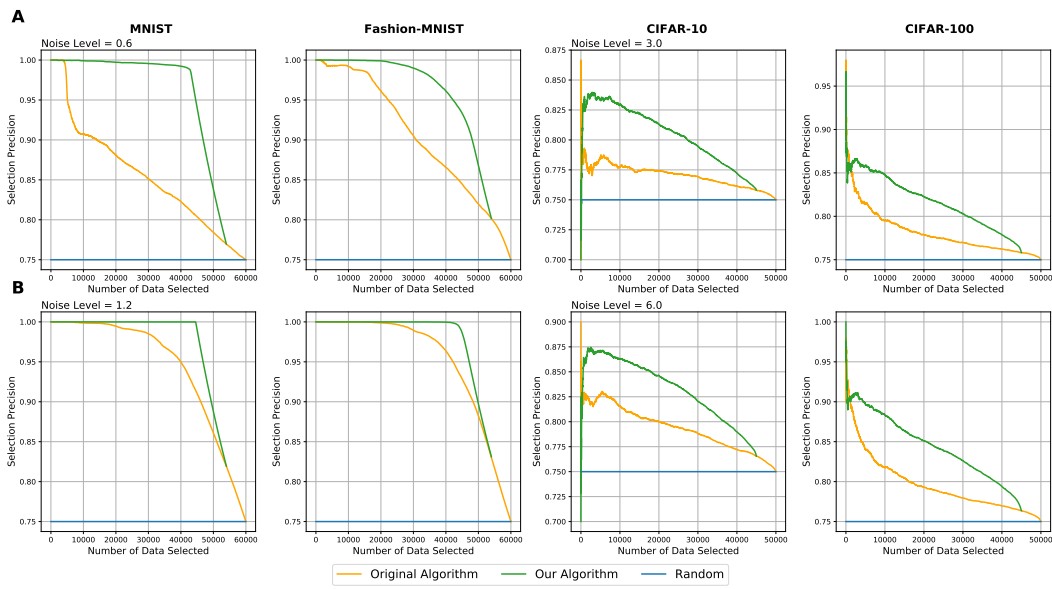

Figure 3: JST framework performance comparison on high-quality data selection, instantiated with optimal transport based method. JST framework shows a notable improvement in selection precision compared with pure data valuation method.

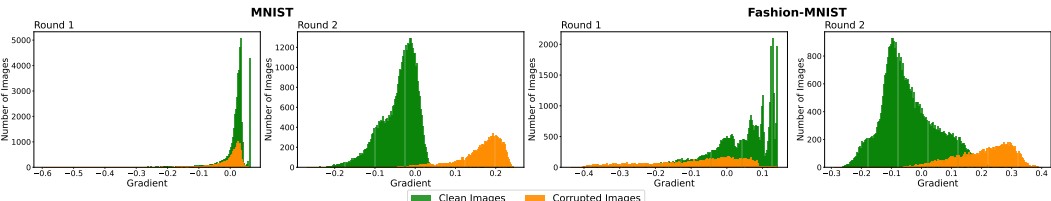

Figure 4: Illustration of the principle of JST framework. In the second-round data selection, high-quality data are popped out as outliers with low value scores, demonstrating near-perfect separation.

## 5 EXPERIMENTAL EVALUATION

In this section, we conduct experiments to verify the effectiveness of our proposed JST framework in augmenting data valuation methods for selecting data that align with the distribution of validation set from training set. We focus on two main tasks: high-quality data selection (Section 5.1) and raw dataset pruning (Section 5.2), using the JST framework instantiated with the optimal transport based data valuation method. Additionally, we demonstrate the general applicability of our JST framework to other data valuation methods, including influence function based and reinforcement learning based approaches (Section 5.4). To cover the complexity of tasks across different scales, we utilize several datasets, including MNIST, Fashion-MNIST (Xiao et al., 2017), CIFAR-10, CIFAR-100 (Krizhevsky et al., 2009), SVHN (Netzer et al., 2011), and Food-101 (Bossard et al., 2014).

### 5.1 HIGH-QUALITY DATA SELECTION

In real-world scenarios, training data often contain corrupted images (Hendrycks & Dietterich, 2019; Li et al., 2020). We evaluate the effectiveness of JST framework in selecting high-quality data in the presence of corrupted images.

**Setup.** We employ four standard datasets: MNIST, Fashion-MNIST, CIFAR-10, and CIFAR-100. For all four datasets, we inject white noise into 25% of the training set and utilize the clean test set as the validation set. To demonstrate the superior performance of JST framework compared with the pure data valuation method, we add two levels of small noise to the training set as depicted in Figure 3 for noise-level comparative analysis. To achieve the goal of identifying "high-quality" training samples, we rank data points based on their value scores in descending order, selecting a subset with the highest values. For every selection budget, we compute the selection precision, i.e, the percentage of the points that are uncorrupted within the selected points. We compare our framework with pure data valuation method and random selection baseline across these four datasets.

**Results.** As depicted in Figure 3, our experimental results show a notable enhancement in selection precision across all four datasets, where our JST framework outperforms the pure data valuation method on nearly all selection budgets. Globally, our framework significantly improves the mean rank of uncorrupted data compared to pure data valuation methods. With a very small noise level of standard deviation 0.6 and 3.0 (Figure 3 A), our framework push forward the mean rank by 3,487 for MNIST, 1,798 for Fashion-MNIST, 700 for CIFAR-10, and 861 for CIFAR-100, from initial values of 24,814, 24,269, 22,212, and 22,106, respectively. This underscores the feasibility of employing data selection using our framework under challenging circumstances with small noise. When facing relatively larger noise with a standard deviation of 1.2 and 6.0 (Figure 3 B), our framework still achieves improvements of 785 for MNIST, 607 for Fashion-MNIST, 843 for CIFAR-10, and 1,167 for CIFAR-100, from initial values of 23,064, 23,072, 21,869, and 21,937, respectively. Remarkably, as illustrated in Figure 4, with a noise level of standard deviation 0.6, the data valuation method after being augmented by our framework achieves near-perfect separation between uncorrupted and corrupted data in the simple datasets MNIST and Fashion-MNIST. Furthermore, Figure 4 clearly confirms the feasibility of the principle of our framework, wherein high-quality data are assigned low value scores and popped out as outliers in the second-round data selection, vice versa to the first-round data valuation.

### 5.2 RAW DATASET PRUNING

With the help of commercial search engines, several web-crawled large datasets have been curated (Li et al., 2017; Sun et al., 2017). These datasets typically contain heterogeneous noise, such as label noise and out-of-distribution samples, and individual training data points usually vary in quality. We further demonstrate the effectiveness of our JST framework in real-world data collection scenarios.

**Setup.** We conduct experiments on two web-crawled datasets, SVHN and Food-101, focusing on the task of raw dataset pruning with being provided a small size of manually cleaned validation set.

More specifically, SVHN is a digit classification dataset with 10 categories (from 0 to 9), cropped from pictures of real-world house number plates obtained from Google Street View images. It contains 73,257 digits for training and 26,032 digits for testing. Similarly, but with greater complexity, Food-101 dataset, which was crawled online, consists of 101 food categories with 750 training images and

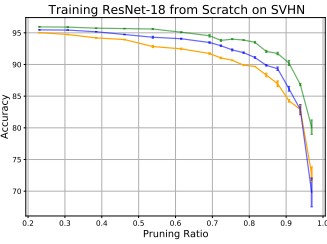 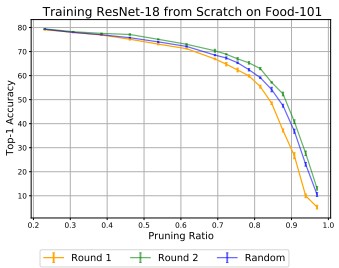 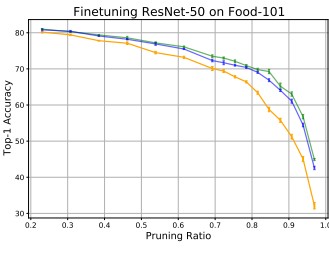

Figure 5: JST framework performance comparison on raw dataset pruning with different pruning ratio, instantiated with optimal transport based method.

250 test images per category, totaling 101,000 images. The labels for test images have been manually cleaned, while the training set contains some noise, primarily in the form of intense colors, incorrect labels and out-of-distribution samples. We randomly sampled 5,000 digits from the test set of SVHN and 7,500 images from the test set of Food-101 to create their respective validation sets, while leaving the remaining samples to be used as the test sets.

Likewise to Section 5.1, we rank training data points based on their value scores in descending order to identify "high-quality" samples. However, to mitigate the issue of class imbalance, which negatively impacts the trained model performance (Johnson & Khoshgoftaar, 2019), we simply rearrange the data points sequentially according to their class one by one after each round of ranking. Then, to evaluate the efficacy of our JST framework, we train a ResNet-18 model from scratch on the training set of SVHN and Food-101 with varying pruning ratios, respectively. Additionally, to explore the effectiveness of our framework on pretrained models, we finetune a ResNet-50 model (He et al., 2016), initially pretrained on ImageNet1K, with different pruning ratios. We test all trained neural networks on the reconstructed testing set, excluding samples from the validation set to prevent prior knowledge and maintain a fair accuracy comparison. Still, we compare our JST framework with the pure data valuation method and random selection baseline.

For the implementation of neural network training, all experiments are conducted on NVIDIA H100 GPUs with PyTorch (Paszke et al., 2019). In training from scratch experiments, we utilize an SGD optimizer with a momentum of 0.9, weight decay of 5e-4, and an initial learning rate of 0.1 with cosine annealing learning rate decay

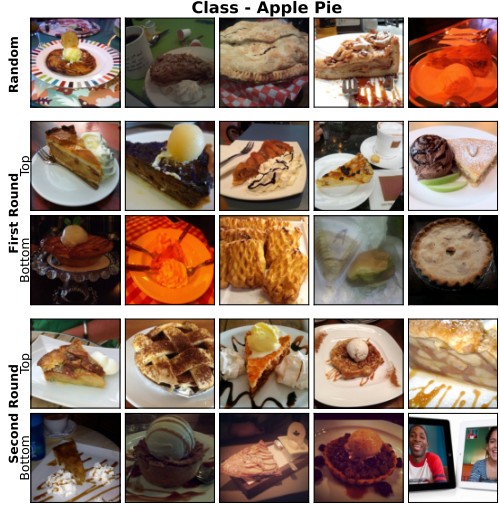

Figure 6: Visualization of extremal images for class apple pie in the Food-101 dataset. The top five (most valuable) and bottom five (least valuable) images are selected. For random selection, after shuffling, first five images are shown. Our framework, compared to the pure data valuation and random baseline, selects diverse, relevant images and excludes low-quality, out-of-distribution samples.

strategy over 200 epochs. For experiments on SVHN, we set a batch size of 256; for experiments on Food-101, we set a batch size of 128 and employ data augmentations of random crop and random horizontal flip. In finetuning experiments on Food-101, the experiment setting is the same as training from scratch experiments except the number of epochs, which is set to 40. Across each experiment, we perform three individual runs with different random seeds and report the mean performance and the standard deviation as error bars.

**Results.** As illustrated in Figure 5, in both SVHN and Food-101 experiments, our framework achieves much higher accuracy compared to the pure data valuation method. It also outperforms the

random selection baseline across most pruning ratios especially in cases with large pruning ratios. Further, in Figure 6, we visualize extremal images of the Food-101 dataset for one class (apple pie) in the ranking. It demonstrates the efficacy of JST, as our framework selects the most relevant images to the class, while random selection and the pure data valuation method include monotone image patterns, low-quality data, or out-of-distribution samples, thereby hurting model performance. More extremal images are deferred at Appendix F.

## 5.3 CROSS-DOMAIN RETRIEVAL

In real-world scenarios, datasets often consist of samples from multiple domains with imbalanced proportions. Cross-domain retrieval aims to identify and select relevant samples from such a mixed-domain dataset to align with the distribution of a given target domain.

**Setup.** We conduct experiments on one time-series tabular dataset: HHAR (Stisen et al., 2015). HHAR dataset is designed to predict human activities within specific time segments, classifying them into six categories: biking, sitting, standing, walking, walking upstairs, and walking downstairs.

Following the data preprocessing method outlined in previous studies (Ragab et al., 2023; He et al., 2023), HHAR dataset is segmented into non-overlapping segments of 128 time steps for classification and then split into training and testing sets as applied in prior research. We further divide HHAR sequential data segments into four domains based on the devices and motion sensors used: phone-accelerometer, watch-accelerometer, phone-gyroscope, and watch-gyroscope. To comprehensively evaluate the effectiveness of our JST framework for cross-domain retrieval, we conduct experiments using each of these domains as the validation set, extracted from the testing set. The training set is treated as a mixed-domain dataset.

Based on previous studies (Kwon & Zou, 2021; Wang & Jia, 2023; Kwon & Zou, 2023), we apply the k-means clustering algorithm to the data values. This approach allows us to partition the training set into two groups: a potential target domain data group and an other data group. Given our expectation that potential target domain data are likely to have high values, we identify the cluster with the higher average data value as the potential target domain data group for retrieval. Following common practice in the literature, we compare the clustering results against the annotated ground truth and calculate the F1-score to evaluate our JST framework. We compare our JST framework with the pure data valuation method. Note that for a fair comparison, we calculate the F1-score using only the data propagated into the second round of our JST framework for the pure data valuation method.

| F1-score | Round 1 | Round 2 |
|---|---|---|
| Phone - Accelerometer | 0.458 ± 0.001 | **0.652 ± 0.001** |
| Watch - Accelerometer | 0.212 ± 0.001 | **0.305 ± 0.001** |
| Phone - Gyroscope | 0.347 ± 0.001 | **0.582 ± 0.002** |
| Watch - Gyroscop | 0.511 ± 0.003 | **0.688 ± 0.001** |

Table 1: JST framework performance comparison on cross-domain retrieval, instantiated with optimal transport based method. The average and standard error of the F1-score are denoted by 'average±standard error'. All the results are based on 10 repetitions. Boldface numbers denote the best method. JST framework shows a notable improvement in F1-score compared with pure data valuation method.

**Results.** As shown in Table 1, our JST framework consistently outperforms the pure data valuation method, even under conditions of severe domain imbalance. Specifically, in the HHAR dataset, the balance score — calculated as the average proportion of the minority domain to the majority domain when selecting two random domains — is as low as 47%. On average, our JST framework improves the F1-score by 0.175 across four domains in the HHAR dataset.

## 5.4 APPLYING JST FRAMEWORK TO DIFFERENT DATA VALUATION METHODS

To demonstrate the general applicability of our JST framework, we apply it to other two data valuation methods: influence function based and reinforcement learning based approaches. We use dataset CIFAR-10 and CIFAR-100 to evaluate the effectiveness of JST. The experimental setup is the same as

in Section 5.1 with white noise standard deviation of 6.0 and 9.0, except the size of training set and validation set. As these two methods suffer from running time for large dataset, to save computational resources, we randomly sample 10,000 data points from the training set as the training set and 1,000 data points from the test data as the validation set. We use the last layer of ResNet-18 model as their dependent training model. As illustrated in Figure 7, these two methods augmented by our framework, JST, improve selection precision to varying extents, corresponding to higher sensitivity of outliers compared with in-distribution data as shown in Figure 1.

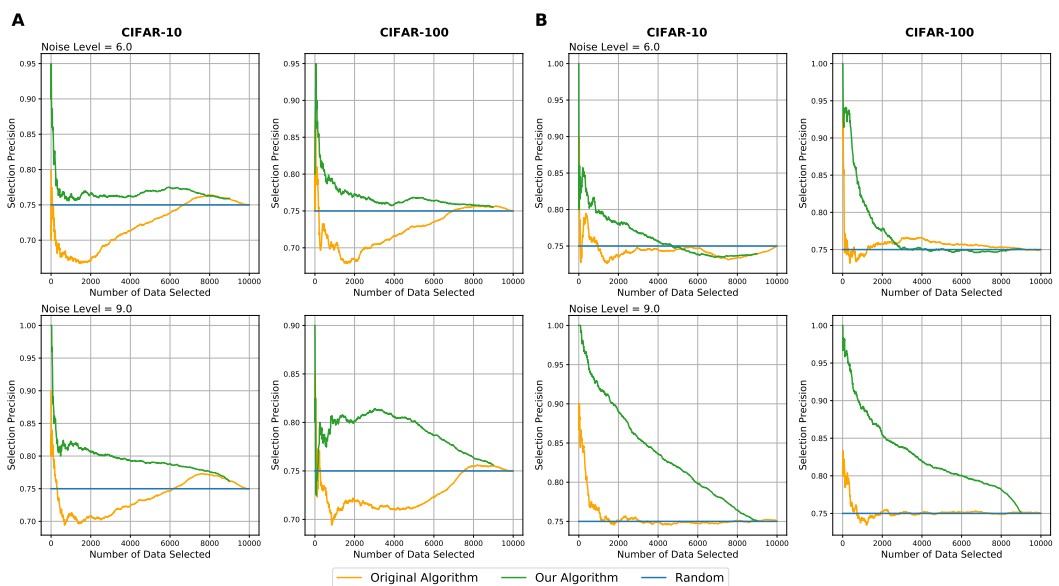

Figure 7: Applying JST framework to different data valuation methods on CIFAR-10 and CIFAR-100. Both methods (**A:** influence function based, **B:** reinforcement learning based) augmented by JST framework improve selection precision to varying extents.

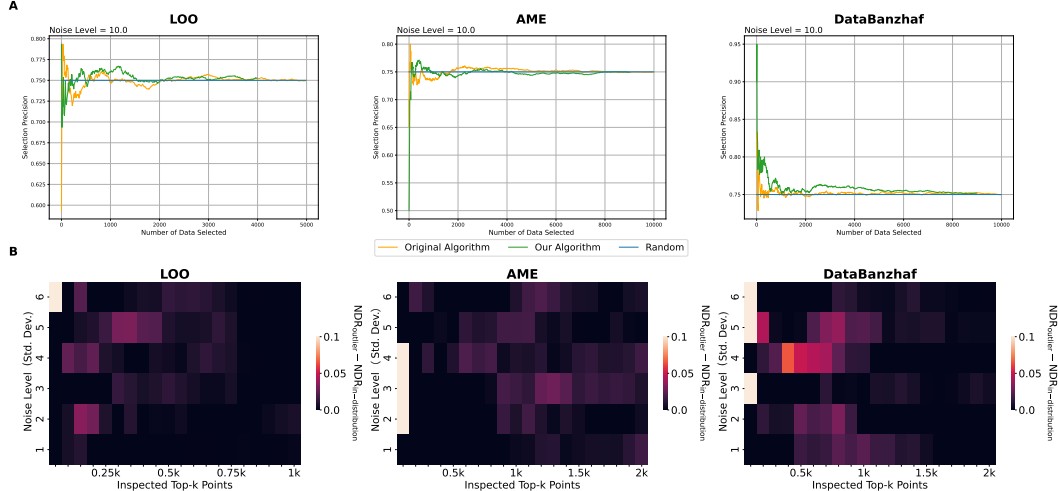

Figure 8: **A:** Evaluating JST framework on marginal contribution based data valuation methods on CIFAR-10. These methods do not improve selection precision compared with the pure data valuation method. **B:** Sensitivity to outliers compared with in-distribution data for marginal contribution based data valuation methods. Following the experimental setup from Figure 1, we evaluate the difference between $NDR_{outlier}$ and $NDR_{in-distribution}$ on CIFAR-10. The results show that these methods exhibit similar levels of sensitivity to outliers and in-distribution data, which explains the underperformance of our approach when applied to marginal contribution based methods.

# 6 WHERE DOES THE JST FRAMEWORK UNDERPERFORM?

Finally, we discuss scenarios where our JST framework underperforms, showing a similar selection precision to the pure data valuation method. Specifically, we observe that marginal contribution based data valuation methods yield unsatisfactory results, as these methods fail to demonstrate a higher detection rate for outliers compared to in-distribution data. We test our framework on three marginal contribution based methods: LOO (Jiang et al., 2023), AME (Lin et al., 2022), and Data Banzhaf (Wang & Jia, 2023). The experimental setup is the same as in Section 5.4 on CIFAR-10 with white noise standard deviation of 10.0, except the reduced size of training set (5,000) and validation set (500) for LOO to save runtime. As shown in Figure 8 A, these three methods augmented by our framework do not improve selection precision. This is because these methods lack the property of higher sensitivity to outliers (Figure 8 B). Based on this underperformance analysis, we propose a simple test framework to validate the expected behavior of our approach: before applying it to real-world applications, we can assess whether the data valuation method exhibits higher sensitivity to outliers on a specific type of data.

# 7 CONCLUSION

In this paper, we introduce JST, a straightforward two-stage framework designed to enhance the sensitivity of existing data valuation methods to high-quality data for subset selection. The JST framework identifies low-quality data points as a validation set, thereby allowing high-quality data to stand out as outliers during the data valuation process. Experiments on multiple datasets demonstrate the effectiveness of JST framework in selecting high-quality data and pruning raw datasets, particularly in scenarios with small noise. We successfully apply JST to different data valuation methods, highlighting the general applicability of our framework, thus making it a valuable tool for enhancing data selection in machine learning.

However, our work focuses solely on data valuation and selection based on loss and accuracy, without considering other important aspects such as fairness across subpopulations, which is crucial in data-centric methods. Additionally, adapting our framework to a wider range of data valuation methods and the currently heated LLM model settings remains a practical interest. We leave these two aspects for future work.

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

APPENDIX

# A    DATA VALUATION ALGORITHMS

This section provides a detailed explanation of three data valuation algorithms applied in our study: optimal transport based method, influence function based method and reinforcement learning based method. Although we have defined notations in the preliminaries section, a thorough set of notations is presented here for clarity. The input space is denoted by $\mathcal{X}$ and the output space by $\mathcal{Y}$. We denote the training set by $D_{tr} = \{z_i\}_{i=1}^n$, where each $z_i = (x_i, y_i)$ is drawn from a source distribution $p_{src}(z)$, the validation dataset by $D_v = \{z_i'\}_{i=1}^m$, where each $z_i' = (x_i', y_i')$ is drawn from a target distribution $p_{trg}(z')$, and a model performance metric *PERF* evaluated on the validation set $D_v$. Typically, $n > m$ and $p_{src}(z)$ is not required to be the same as $p_{trg}(z')$, i.e., $p_{src}(z) \neq p_{trg}(z')$. The goal of data valuation is to distribute the validation performance across training data points and compute a data score value $s(z_i)$ for each training data point $z_i$.

LAVA (OPTIMAL TRANSPORT BASED)

LAVA (Just et al., 2023) measures the sensitivity of validation performance to changes in the training data. It examines how the optimal transport cost between the training set $D_{tr}$ and the validation set $D_v$ changes when a particular data point in $D_{tr}$ is assigned increased weight. The sensitivity is determined by calculating the gradient of the optimal transport cost with respect to the probability mass.

The optimal transport cost gradient can be calculated as the data value score for the training data point $z_i$ as follows:

$$s(z_i) := t^*[i] - \frac{1}{n-1} \sum_{j \neq i} (t^*[j])$$

where $\{t_i^*\}_{i=1}^n$ is the optimal solution of the dual problem, which is expressed as:

$$t^*, u^* := \arg \max_{t,u \in \mathcal{C}^0(\mathcal{X} \times \mathcal{Y})^2} \left\langle t, \frac{1}{n} \delta_{(x_i, y_i)} \right\rangle + \left\langle u, \frac{1}{m} \delta_{(x_i', y_i')} \right\rangle$$

where $\mathcal{C}^0(\mathcal{X} \times \mathcal{Y})$ denotes the set of all continuous functions defined on $\mathcal{X} \times \mathcal{Y}$, and $\delta_{(x,y)}$ represents the Delta measure at $(x, y) \in \mathcal{X} \times \mathcal{Y}$. The negative gradient indicates that the loss will decrease when the data point is given more weight, signifying a higher value for the data point, whereas a positive gradient indicates the opposite.

INFLUENCE FUNCTION

In machine learning, influence function (Koh & Liang, 2017) is used to evaluate the impact of a data point on a model's performance by upweighting a specific training point. We denote the influence of a training data point $z_i = (x_i, y_i)$ on the loss $L(z, \theta)$ with respect to the parameters $\theta \in \Theta$ of a validation data point $z_j' = (x_j', y_j')$ as $I(z_i, z_j')$. This can be obtained by:

$$I(z_i, z_j') := -\nabla_\theta L(z_j', \hat{\theta})^\top \mathcal{H}_{\hat{\theta}}^{-1} \nabla_\theta L(z_i, \hat{\theta})$$

where $\nabla_\theta L(z_j', \hat{\theta})$ represents the gradient of the loss $L(z_j', \hat{\theta})$ with respect to optimized parameters $\hat{\theta}$ evaluated at the validation data point $z_j'$. Similarly, $\nabla_\theta L(z_i, \hat{\theta})$ represents the gradient of the loss $L(z_i, \hat{\theta})$ with respect to optimized parameters $\hat{\theta}$ evaluated at the training data point $z_i$. The term $\mathcal{H}_{\hat{\theta}}$ denotes the Hessian matrix of empirical risk with respect to the optimized model parameters $\hat{\theta}$, defined as $\frac{1}{n} \sum_{i=1}^n \nabla_\theta^2 L(z_i, \hat{\theta})$.

The negative influence predicts a decrease in loss, while the positive influence predicts an increase. Therefore, higher negative influences correspond to more valuable data points, whereas larger positive influences correspond to less valuable points. To evaluate the impact of each training data point $z_i$

on the whole valuation set $D_v$, we simply sum the influence of the training data point $z_i$ on each validation data point $z'_j$ as the data value score $s(z_i)$, denoted as:

$$s(z_i) := \sum_{j=1}^{m} I(z_i, z'_j)$$

DVRL (REINFORCEMENT LEARNING BASED)

DVRL (Yoon et al., 2020) involves using reinforcement learning algorithm to compute the importance weight for each training data point as the data value score. The objective function that DVRL solves in training a model $g : \mathcal{X} \times \mathcal{Y} \to [0, 1]$, which maps data points to their importance weight, is expressed as:

$$\min_{g \in G} \mathbb{E}_{(x', y') \sim p_{trg}(z')} [\mathbb{L}(f_g(x'), y')]$$

$$\text{s.t.} \quad f_g = \arg \min_{f \in F} \mathbb{E}_{(x, y) \sim p_{src}(z)} [g(x, y) \cdot \mathbb{L}(f(x), y)]$$

where $G := \{g : \mathcal{X} \times \mathcal{Y} \to [0, 1]\}$, $F := \{f : \mathcal{X} \to \mathcal{Y}\}$ and the loss $\mathbb{L}$ can be MSE or cross entropy. The data value score of a training data point $s(z_i)$ is computed as the importance weight $g(x_i, y_i)$. The objective can be optimized using policy gradient methods. A large importance weight indicates a more crucial data point for the training process, signifying its high value.

## B  PRACTICAL IMPLEMENTATION OF JST

This section we provide the ablation studies of JST framework instantiated by optimal transport based data valuation method on CIFAR-10 dataset with a noise level of standard deviation 6.0.

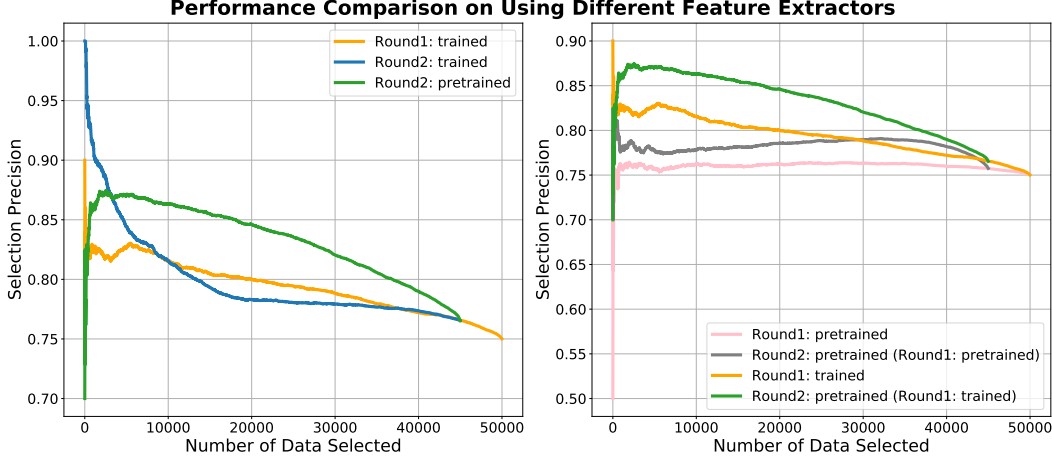

Figure 9: Performance comparison on using different feature extractors in each round. **A:** In the second-round data selection, we have ResNet-18 model trained on the validation set $D_v$ to extract features compared with a pretrained ResNet-18 model on ImageNet1K. The feature extractor of ResNet-18 model trained on the validation set $D_v$ hurts the performance in the second-round data selection. **B:** In both rounds, we have pretrained ResNet-18 model on ImageNet1K to extract features. Our JST framework improves selection precision, though the improvement is less pronounced in the second-round, compared with employing ResNet-18 model trained on the validation set $D_v$ in the first-round data valuation. This confirms that the improvement in our JST framework is not due to switching to a pretrained ResNet-18 model on ImageNet1K for the second-round data selection.

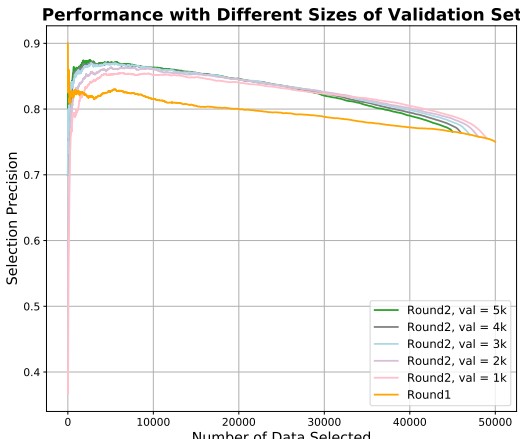

Figure 10: Performance comparison on using different sizes of validation set in the second-round data selection. In the second-round, we ablate different sizes of validation set $|D_{sel}|$ to use, including 5,000, 4,000, 3,000, 2,000, 1,000. The selection precision lines closely align together, indicating the robustness of our JST framework, not requiring many signals from low-quality data to pop out high-quality data.

## C HOW TO COMPUTE SENSITIVITY COMPARISON?

For in-distribution data detection rate, we rank data points based on their value scores in descending order, selecting a subset with the highest values. For every selection budget, we compute the selection precision, i.e, the percentage of the points that are uncorrupted within the selected points, as in-distribution data detection rate. For outlier detection rate, we rank data points based on their value scores in ascending order, selecting a subset with the lowest values. For every selection budget, we compute the selection precision, i.e, the percentage of the points that are corrupted within the selected points, as outlier detection rate. Finally, to account for potential imbalance between corrupted and uncorrupted data, we normalize both detection rates.

## D FURTHER VISUALIZATION OF RAW DATASET PRUNING

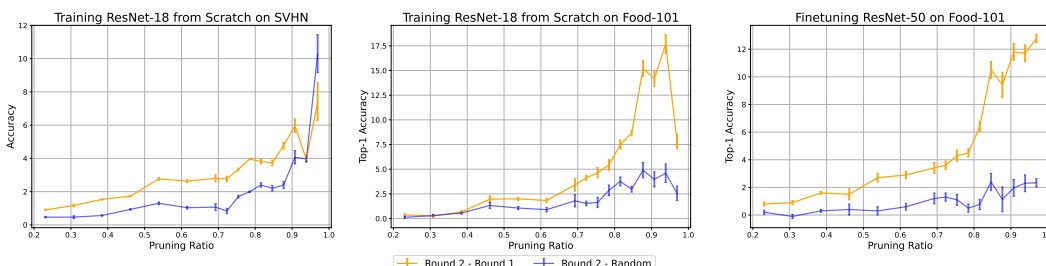

Figure 11: Further visualization of JST framework performance comparison on raw dataset pruning. We plot the differences on accuracy between our framework and both the pure data valuation method and the random selection baseline, highlighting the improvements achieved by JST framework.

# E  RUNTIME COMPARISON

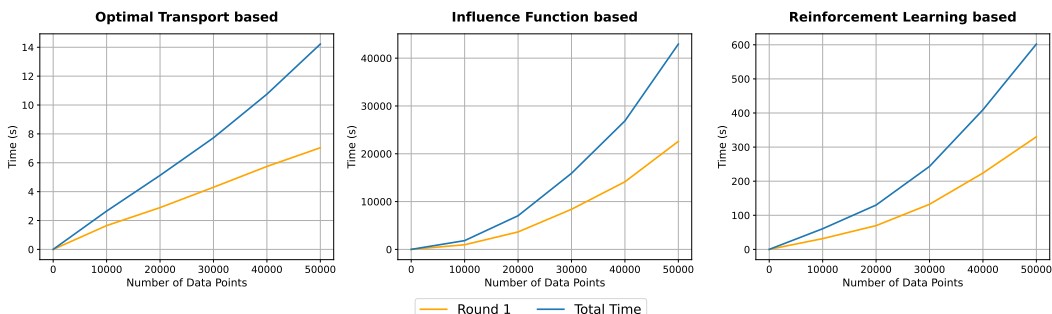

Figure 12: Runtime comparison on CIFAR-10 with varying sizes of training set. We use the same experimental setup as described in Section 5.1 but vary the dataset size. We observe that our JST framework approximately doubles the runtime across all methods, as it does not require additional overhead. The optimal transport based method stands out for its exceptional efficiency, completing in just a few seconds while exhibiting favorable near-linear time complexity (Just et al., 2023).

| JST on Datasets | Time |
| --- | --- |
| SVHN - Round 1 | 9.9 sec. |
| SVHN - Total | 21.6 sec. |
| Food101 - Round 1 | 44.4 sec. |
| Food101 - Total | 90.7 sec. |

Table 2: Runtime comparison for the raw dataset pruning task. Leveraging the exceptional efficiency of the optimal transport based method, our JST framework achieves data valuation in remarkably short time.

# F  EXTREMAL IMAGES

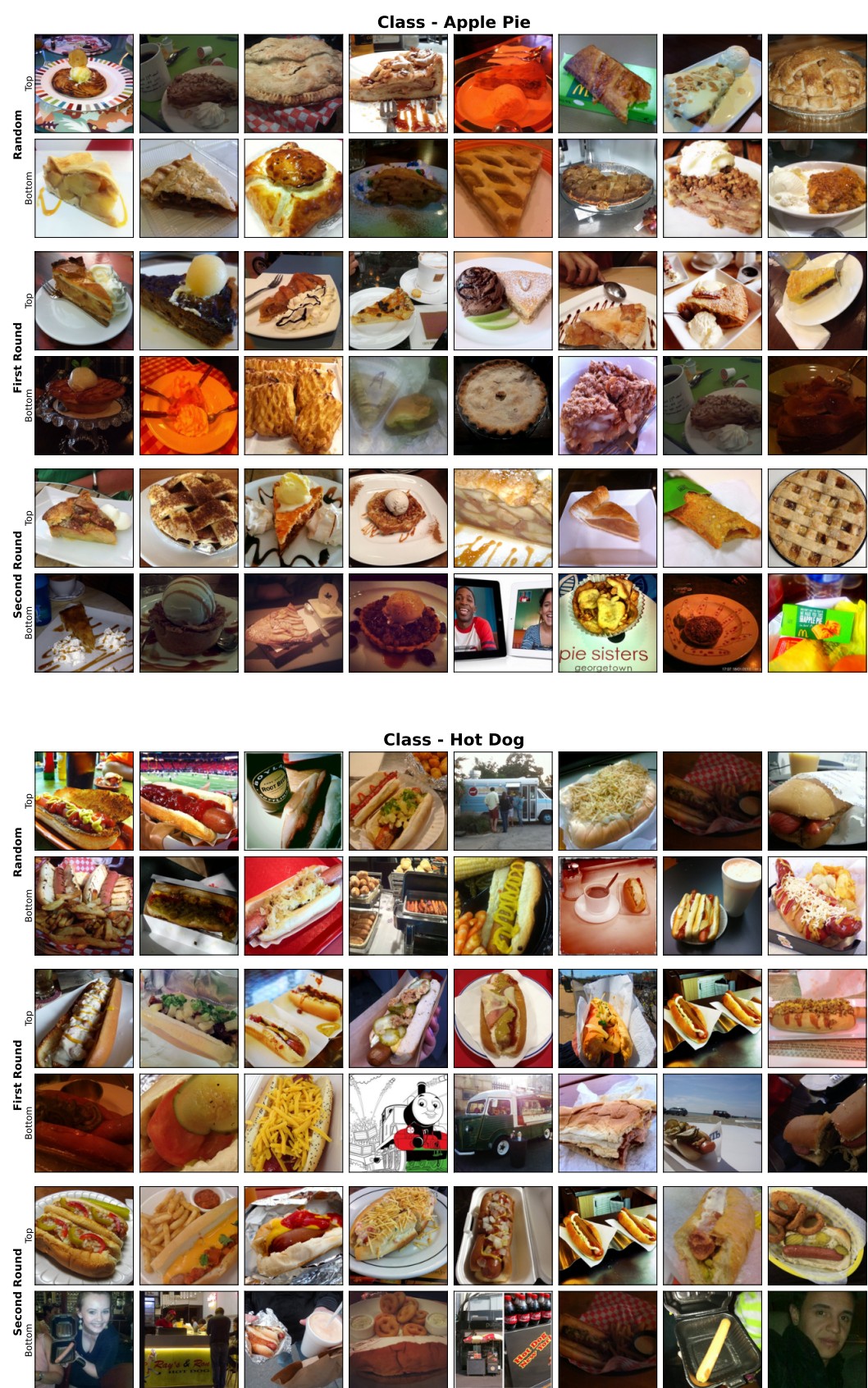

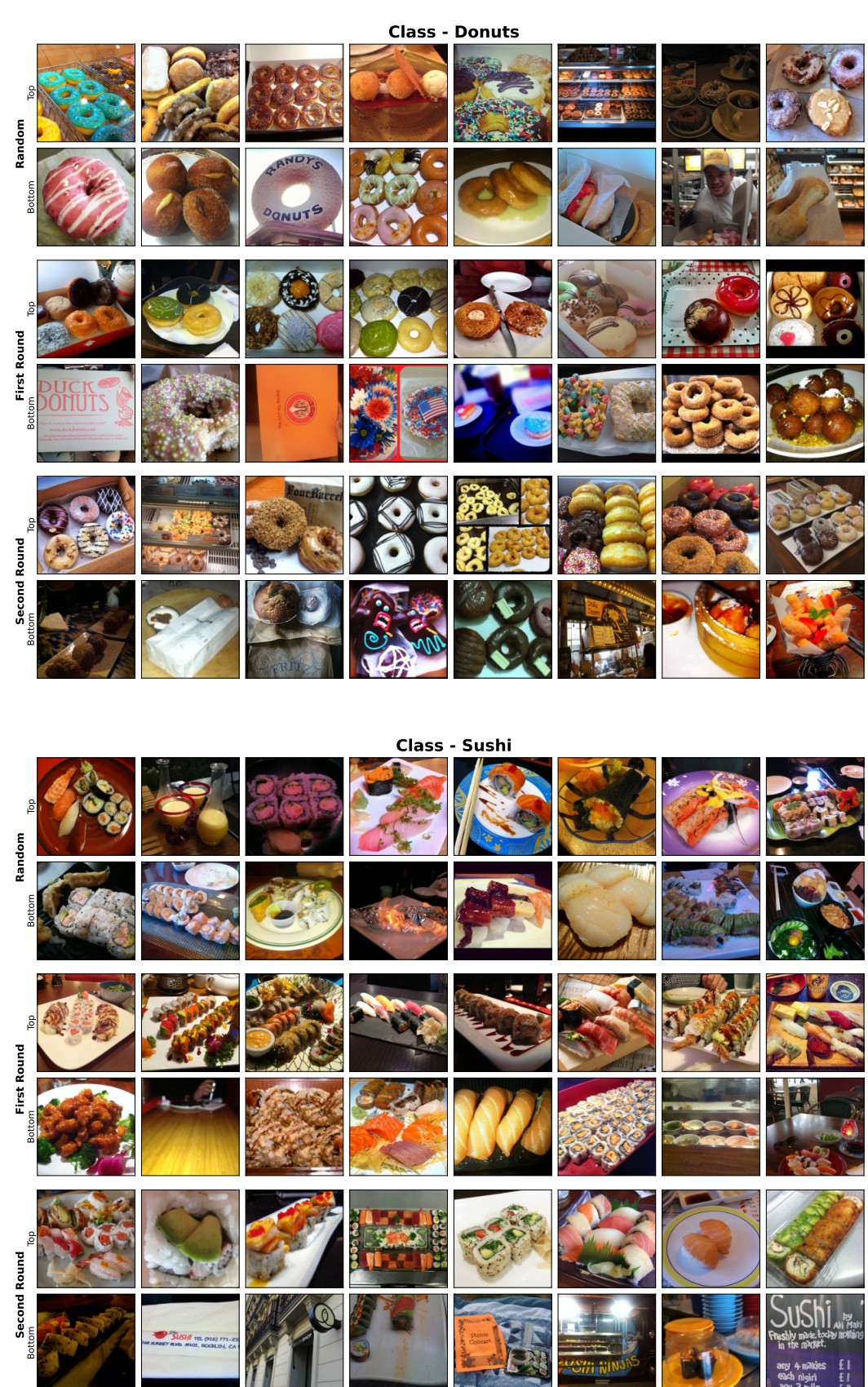

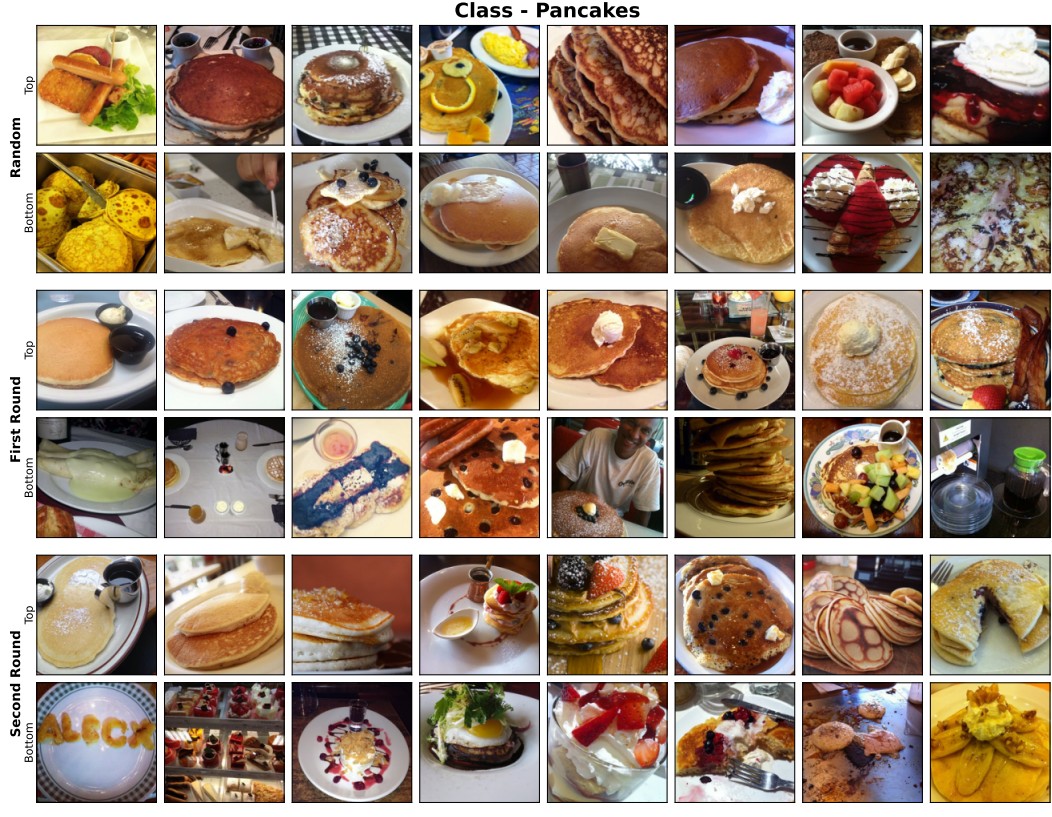

**Class - Pancakes**