# OpenReview forum: "Just Select Twice: Leveraging Low Quality Data to Improve Data Selection"
_ICLR.cc/2025/Conference — Submitted to ICLR 2025_

### Official Review · Reviewer_CgxJ · 2024-10-29

**Soundness:** 2
**Presentation:** 2
**Contribution:** 2
**Rating:** 3
**Confidence:** 4

**Summary:**

The paper proposes to enhance data selection by performing a second round of data valuation. The first round of data valuation is done using a validation set. The proposed algorithm would identify $k$ points with the lowest valuation (i.e., performance on the validation set). The second round would value the remaining points based on how poorly they perform on this low value set. Points that perform poorly are high quality data.

**Strengths:**

It is important to study how data valuation can be better adapted to select high quality data (instead of only filtering out low quality data). The experiments consider various datasets and data valuation methods.

**Weaknesses:**

1. The clarity of the paper can be improved. The paper repeatedly use the word "outlier" but does not define or use the word precisely. How do we distinguish between outliers and in-distribution data? How do outliers differ from noisy data? Does the classification depend on the data valuation method? Without clear examples or definition, the significance of Figure 1 is unclear.  The definition of outlier seems to change in line 79 "allows high-quality data to be identified as outliers in the new context". The authors can provide a clear definition or example at the start of the paper and use it consistently.
2. The experiments only consider corrupted images. Would the JST algorithm perform well for other scenarios such as corrupted labels? For example, the first round of data valuation might select some data with wrong label A. In the second round, any data with different label (e.g., true label B and wrong label C) would have the same low score. The revision should. include more experiments to prove the method is useful for other data corruption.

Minor comments: Citations should be in the form of (author, year) instead of author (year).

**Questions:**

1. Explain how the normalised outlier detection and in-distribution data detection rate is computed in Fig 1. Is there some ground truth to classify whether data is an outlier or in-distribution?
Ideally, how should Fig 1 look like for a good data valuation method? Is that achieved by your method?
2. What is a more precise definition for sensitivity to outliers and sensitivity to high quality data?
3. Would the JST algorithm work for tabular data and other form of errors such as corrupted labels?

---

> ### Author Response · Authors · 2024-12-02
> **The clarity of the paper can be improved.**
>
> > *The clarity of the paper can be improved. The paper repeatedly use the word "outlier" but does not define or use the word precisely. How do we distinguish between outliers and in-distribution data? How do outliers differ from noisy data? Does the classification depend on the data valuation method? Without clear examples or definition, the significance of Figure 1 is unclear. The definition of outlier seems to change in line 79 "allows high-quality data to be identified as outliers in the new context". The authors can provide a clear definition or example at the start of the paper and use it consistently.*
>
> Re:
> In a general data valuation framework, a clean and high-quality validation set is typically provided and the goal is to assess the contribution of each individual training data points to the overall performance on this provided validation set. Therefore, data valuation formalizes data selection by identifying a subset of training set to include or exclude based on alignment with the target distribution represented by the validation set, incorporating validation performance into individual training samples [1, 2].
>
> So, outliers or in-distribution data are determined with respect to the validation set provided, regardless of data valuation methods used. In typical data valuation methods, a clean validation set is provided, so the noisy data is treated as outliers. In contrast, we augment existing data valuation methods in a reverse way by leveraging the detected noisy data as the validation set in the second-round of our framework. The validation set is noisy, so high-quality clean data are considered as outliers. We have clearly discussed the goal of data valuation and data selection in the related work and formally defined the framework of data valuation and data selection in the preliminary.
>
> To clarify further, we have added a footnote of an explanation in the abstract. For the line 79, we have updated it to “Our key insight is that incorporating a second-round subset selection, using detected outliers **as the new validation set**, allows high-quality data to be identified as outliers **with respect to the new validation set in this new context**.”
>
> > [1] Just, H. A., Kang, F., Wang, T., Zeng, Y., Ko, M., Jin, M., & Jia, R. (2023, February). LAVA: Data Valuation without Pre-Specified Learning Algorithms. In *The Eleventh International Conference on Learning Representations*. OpenReview. \
> > [2] Yoon, J., Arik, S., & Pfister, T. (2020, November). Data valuation using reinforcement learning. In *International Conference on Machine Learning* (pp. 10842-10851). PMLR.

---

> > ### Comment · Reviewer_CgxJ · 2024-12-03
> > **Outlier**
> >
> > I am looking for mathematical or algorithm-based definitions. For example, are outliers just the points with the lowest data values?

---

> ### Author Response · Authors · 2024-12-02
> **Citations should be in the form of (author, year) instead of author (year).**
>
> We apologize for the citation format error and have corrected it in the updated version of the paper.

---

> ### Author Response · Authors · 2024-12-02
> **The experiments only consider corrupted images.**
>
> >*The experiments only consider corrupted images. Would the JST algorithm perform well for other scenarios such as corrupted labels? For example, the first round of data valuation might select some data with wrong label A. In the second round, any data with different label (e.g., true label B and wrong label C) would have the same low score. The revision should. include more experiments to prove the method is useful for other data corruption.*
>
> Re: Our objective is to augment existing data valuation methods, particularly under conditions of mild noise. To this end, we primarily consider corrupted images as corrupted images present more challenging scenarios for the task of leveraging data valuation in data selection compared with the strong noise type of corrupted labels. Furthermore, in real-world raw dataset pruning, we consider more complex usage cases containing heterogeneous noise with variations in image quality, including incorrect labels and out-of-distribution samples.
>
> As demonstrated in Section A (https://anonymous.4open.science/r/JST-005F/JST_main_track_appendix_reply.pdf), our algorithm performs well in cases of corrupted labels. For this evaluation, we use a standard approach of randomly label-flipping with a 25% corruption ratio on dataset CIFAR-10 and employ optimal transport based, reinforcement learning based and influence function based data valuation methods. Our framework still outperforms pure data valuation methods to varying extents. The reason is that in the first round, it is likely to capture a general distribution of wrong labels as the new validation set if not the case that one type of wrong label is very rare. Then in the second-round of selection, opposed by the new validation set, not likely true label and one type of wrong label have the same low score.
>
> To further investigate the interaction effect of different types of corruption, we evaluate our framework on a mixed setting on CIFAR-10 comprising 12.5% feature noise (maintaining the consistent noise level described in the paper) and 12.5% label noise. As demonstrated in Section B (https://anonymous.4open.science/r/JST-005F/JST_main_track_appendix_reply.pdf), our framework continues to outperform pure data valuation methods to varying extents.

---

> ### Author Response · Authors · 2024-12-02
> **Explain how the normalised outlier detection and in-distribution data detection rate is computed in Fig 1.**
>
> >*Explain how the normalised outlier detection and in-distribution data detection rate is computed in Fig 1. Is there some ground truth to classify whether data is an outlier or in-distribution? Ideally, how should Fig 1 look like for a good data valuation method? Is that achieved by your method?*
>
> Re:
> 1. **How to compute normalized outlier detection rate and in-distribution data detection rate?**
>
> To evaluate outlier detection, we rank data points by their value scores in ascending order and select a subset with the lowest scores. The selection precision, defined as the percentage of corrupted points in the subset, represents the outlier detection rate. Similarly, for in-distribution data detection, we rank data points by their value scores in descending order and select a subset with the highest scores. The selection precision, defined as the percentage of uncorrupted points, represents the in-distribution detection rate.
>
> To account for potential imbalance between corrupted and uncorrupted data, we normalize both detection rates. First, we compute the improvement over the precision of random selection baseline. Next, we divide this improvement by the maximum achievable gain, defined as the difference in precision between optimal selection and the random selection baseline. For illustration, consider a training set of 10,000 data points, with 50\% corrupted and 50\% uncorrupted. For a given selection budget, improvements in in-distribution data and outlier detection are measured against the 50\% precision of random selection baseline (i.e., the corrupted and uncorrupted ratios). If the selection budget is 5,000 or less, the optimal selection precision is set to 1.0. If the selection budget exceeds 5,000, the optimal selection precision is computed as the ratio of 5,000 to the selection budget.
>
> Note that we use 10,000 data points as the training set and 1,000 data points as the validation set, randomly sampled from the training and test data of CIFAR-10, respectively. For LOO method, the training set is reduced to 5,000 and the validation set to 500, to save runtime. To visualize the difference between the normalized outlier detection rate (NDR_outlier) and the normalized in-distribution data detection rate (NDR_in-distribution), we inspect the top 20\% of data points, except for influence function based method, where the top 40\% is considered for a complete display.
>
> 2. **Ground truth to classify whether data are outliers or in-distribution data.**
>
> As discussed in the previous question, outliers or in-distribution data are determined with respect to the validation set provided, regardless of data valuation methods used. Figure 1 shows the normalized outlier detection rate (NDR_outlier) against the normalized in-distribution data detection rate (NDR_in-distribution) for original data valuation methods. Here, the validation set of clean data is provided, so corrupted data are outliers and uncorrupted data are in-distribution data.
>
> 3. **How Figure 1 looks like for an improved data valuation method?**
>
> Ideally, augmenting existing data valuation methods with our framework enhances sensitivity to high-quality data, reversing the difference between the normalized outlier detection rate (NDR_outlier) and the normalized in-distribution detection rate (NDR_in-distribution). This results in negative values, where NDR_in-distribution exceeds NDR_outlier. As demonstrated in Section C (https://anonymous.4open.science/r/JST-005F/JST_main_track_appendix_reply.pdf), our framework achieves this ideal outcome.

---

> ### Author Response · Authors · 2024-12-02
> **What is a more precise definition for sensitivity to outliers and sensitivity to high quality data?**
>
> >*What is a more precise definition for sensitivity to outliers and sensitivity to high quality data?*
>
> Re: We define sensitivity to outliers and high-quality data using detection rates. Sensitivity to outliers is measured as the unbiased improvement in outlier detection rate over the random selection baseline, represented by the normalized outlier detection rate (NDR_outlier). Similarly, sensitivity to high-quality data is defined as the unbiased improvement in in-distribution data detection rate over the random selection baseline, represented by the normalized in-distribution data detection rate (NDR_in-distribution).

---

> ### Author Response · Authors · 2024-12-04
> **Would the JST algorithm work for tabular data and other form of errors such as corrupted labels?**
>
> >*Would the JST algorithm work for tabular data and other form of errors such as corrupted labels?*
>
> Re: To cover the complexity of datasets we evaluate on, we have included a time-series tabular dataset HHAR [1] for the task of cross-domain retrieval. Based on prior studies [2, 3], we divide HHAR sequential data into four domains based on devices and motion sensors used, using each domain as the validation set while treating the training set as a mixed-domain dataset. Based on previous studies [4, 5], we apply the k-means clustering algorithm to the data values and partition the training set into two groups. We identify the cluster with the higher average data value as the potential target domain data for retrieval. Then to evaluate our JST framework, we compare clustering results against ground truth and calculate the F1-score. We compare our JST framework with the pure data valuation method. Note that for a fair comparison, we calculate the F1-score using only the data propagated into the second round of our JST framework for the pure data valuation method.
>
> For tabular data type, feature extraction is not required, as noted in prior studies [4, 5]. For influence function based and reinforcement learning based data valuation methods, we follow previous studies [2, 3] to use a 1D-CNN model as the dependent model and to reduce the running time, we randomly sample 10% of data to utilize. The results run independently and repeatedly in 10 times and we report the ’average±standard error’. Boldface numbers denote the best method.
>
> As demonstrated in Section D (https://anonymous.4open.science/r/JST-005F/JST_main_track_appendix_reply.pdf), our JST framework consistently outperforms the pure data valuation method, even under conditions of severe domain imbalance. Specifically, in the HHAR dataset, the balance score — calculated as the average proportion of the minority domain to the majority domain when selecting two random domains — is as low as 47\%.
>
> For the corruption type of labels, we have discussed in the previous question.
>
> >[1] Stisen, A., Blunck, H., Bhattacharya, S., Prentow, T. S., Kjærgaard, M. B., Dey, A., ... & Jensen, M. M. (2015, November). Smart devices are different: Assessing and mitigatingmobile sensing heterogeneities for activity recognition. In *Proceedings of the 13th ACM conference on embedded networked sensor systems* (pp. 127-140).\
> >[2] Ragab, M., Eldele, E., Tan, W. L., Foo, C. S., Chen, Z., Wu, M., ... & Li, X. (2023). Adatime: A benchmarking suite for domain adaptation on time series data. *ACM Transactions on Knowledge Discovery from Data*, *17*(8), 1-18.\
> >[3] He, H., Queen, O., Koker, T., Cuevas, C., Tsiligkaridis, T., & Zitnik, M. (2023, July). Domain adaptation for time series under feature and label shifts. In *International Conference on Machine Learning* (pp. 12746-12774). PMLR.\
> >[4] Kwon, Y., & Zou, J. (2021). Beta shapley: a unified and noise-reduced data valuation framework for machine learning. arXiv preprint arXiv:2110.14049.\
> >[5] Jiang, K., Liang, W., Zou, J. Y., & Kwon, Y. (2023). Opendataval: a unified benchmark for data valuation. Advances in Neural Information Processing Systems, 36.

---

> ### Author Response · Authors · 2024-12-04
>
> Thanks for your reply. We consider corrupted data as the outliers (ground truth) if given a clean validation set but predict the lowest data values as the predicted outliers (some of them might not be the outliers compared with ground truth) to exclude by data valuation methods. We have included more detailed mathematical definitions in our theoretical analysis part.

---

### Official Review · Reviewer_QHSD · 2024-10-30

**Soundness:** 2
**Presentation:** 2
**Contribution:** 2
**Rating:** 3
**Confidence:** 4

**Summary:**

This paper studies data valuation, specifically using data values to perform data selection. The authors proposed to exploit a property of optimal transport-based data valuation methods, namely their sensitivity to outliers (and insensitivity to inliers), and describe a two stage algorithm. The proposed algorithm first selects outliers, and then in the second stage use the selected outliers as validation dataset to again select outliers. Due to the statistical difference, the outliers selected in the second stage are high quality data points. Empirical results on image datasets are provided.

**Strengths:**

- The studied problem is relevant and important.
- The method is described clearly.
- Some empirical validation is provided.

**Weaknesses:**

- There does not seem to be theoretical justification or characterization of the effectiveness of this method. In comparison, the optimal transport based method (Just et al., 2023) or the influence based method (Koh et al., 2017) both provide theoretical justifications.

- The empirical performance can be made more extensive. There are quite a few existing data valuation methods, such as those described in (Sim et al, 2022). While it is understood the proposed method is motivated by the property of optimal transport-based method, the authors indeed investigate other methods such as reinforcement learning based method (Yoon et al., 2020). Thus, it makes sense to compare with these existing methods.

- The key property of this method requires that low-quality and high-quality data to be "outliers" to each other. But in practice, this relationship is not so crisp. For a specific task, the characterization of ideal high-quality data is fairly narrow (i.e., there is only one kind of data which is high quality), while there are (possibly infinitely) many characterizations of low-quality data/outliers. While it is true that outliers (say outlier distribution A) would be statistically different from high quality data (outliers to high quality data), but another type of outlier (say outlier distribution B) would also be outlers to outlier distribution A, but this does not make it high quality. In other words, high quality is the outlier to outliers, but the reverse need not be true (an outlier to an outlier is a high quality data point).

**Questions:**

In abstract,

`Many data valuation methods are sensitive to outliers and require a certain level of noise to effectively distinguish low-quality data from high-quality data,`

What is meant by "require a certain level of noise... "?


In Figure 1, how are the rates computed? and why do they show a higher sensitivity of data valuation methods to outliers?

Instead of ResNet-18 and ResNet-18 pretrained on Image1K, what do you propose as a principled and general way to perform feature extraction for a dataset to use your method?

What would be the empirical results if there are different types of outliers present? For instance two differnt types of noise.

---

> ### Author Response · Authors · 2024-11-26
> **The empirical performance can be made more extensive.**
>
> > *The empirical performance can be made more extensive. There are quite a few existing data valuation methods, such as those described in (Sim et al, 2022). While it is understood the proposed method is motivated by the property of optimal transport-based method, the authors indeed investigate other methods such as reinforcement learning based method (Yoon et al., 2020). Thus, it makes sense to compare with these existing methods.*
>
> Re: Our paper aims to augment data valuation methods to improve their sensitivity to high-quality data for data selection. Therefore, we specifically focus on the field of data valuation. According to the literature [1], data valuation methods can be broadly categorized into three approaches: (1) gradient (optimal transport based and influence function based), (2) importance weight (reinforcement learning based) and marginal contribution (e.g., LOO, DataBanzhaf, AME).
>
> Our proposed JST framework demonstrates strong performance with gradient based and importance weight based methods. Also, we have shown for marginal contribution based methods, our framework underperforms, as these methods fail to demonstrate a higher detection rate for outliers compared to in-distribution data. The only exception is an out-of-bag estimate method [2], which we do not discuss in detail because it does not utilize a validation set as a reference for high-quality data distribution, making it incompatible with our framework. Therefore, our study comprehensively addresses all aspects of data valuation methods, including those described in [3].
>
> > [1] Jiang, K., Liang, W., Zou, J. Y., & Kwon, Y. (2023). Opendataval: a unified benchmark for data valuation. *Advances in Neural Information Processing Systems, 36*. \
> > [2] Kwon, Y., & Zou, J. (2023, July). Data-oob: Out-of-bag estimate as a simple and efficient data value. In *International Conference on Machine Learning* (pp. 18135-18152). PMLR. \
> > [3] Sim, R. H. L., Xu, X., & Low, B. K. H. (2022, July). Data Valuation in Machine Learning: "Ingredients," Strategies, and Open Challenges. In *IJCAI* (pp. 5607-5614).

---

> > ### Comment · Reviewer_QHSD · 2024-11-27
> > **Thank you for the rebuttal**
> >
> > I thank the authors for their response.
> >
> > The authors mention that they
> > > specifically focus on the field of data valuation
> >
> > and demonstrate performance improvements over some existing baselines, but not all, for instance the out-of-bag estimate data valuation method, which the authors mention is
> > > incompatible with our framework.
> >
> > While I can see that there are differences in the setting, it remains to be seen that the proposed method (in this paper) is indeed a better solution to the problem (i.e., data valuation). It does seem that my other questions and comments were not specifically addressed. I would maintain my overall recommendation.

---

> ### Author Response · Authors · 2024-12-03
> **The key property of this method requires that low-quality and high-quality data to be "outliers" to each other.**
>
> >*The key property of this method requires that low-quality and high-quality data to be "outliers" to each other. But in practice, this relationship is not so crisp. For a specific task, the characterization of ideal high-quality data is fairly narrow (i.e., there is only one kind of data which is high quality), while there are (possibly infinitely) many characterizations of low-quality data/outliers. While it is true that outliers (say outlier distribution A) would be statistically different from high quality data (outliers to high quality data), but another type of outlier (say outlier distribution B) would also be outlers to outlier distribution A, but this does not make it high quality. In other words, high quality is the outlier to outliers, but the reverse need not be true (an outlier to an outlier is a high quality data point).*
>
> Re: As demonstrated in Section B (https://anonymous.4open.science/r/JST-005F/JST_main_track_appendix_reply.pdf), our algorithm performs well in cases of being with different types of corruption. We evaluate our framework on a mixed setting on CIFAR-10 comprising 12.5% feature noise (maintaining the consistent noise level described in the paper) and 12.5% label noise (a standard approach of randomly label-flipping) and employ optimal transport based (Subfigure A), influence function based (Subfigure B) and reinforcement learning based (Subfigure C) data valuation methods. Our framework still outperforms pure data valuation methods to varying extents. The reason is that in the first round, it is likely to capture a general distribution of different types of outliers as the new validation set if not the case that one type of outlier is very rare. Then in the second-round of selection, opposed by the new validation set, high-quality data could be popped out.
>
> Furthermore, in real-world raw dataset pruning, we have already considered more complex usage cases containing heterogeneous noise with variations in image quality, including incorrect labels and out-of-distribution samples. Our framework surpasses pure data valuation method and random selection baseline in the downstream tasks of training and finetuning neural networks.

---

> ### Author Response · Authors · 2024-12-03
> **What is meant by "require a certain level of noise... "?**
>
> >*In abstract, "many data valuation methods are sensitive to outliers and require a certain level of noise to effectively distinguish low-quality data from high-quality data," what is meant by "require a certain level of noise... "?*
>
> Re: We apologize for any confusion caused by the wording. We intended to convey that many data valuation methods are sensitive to outliers and require a large level of noise in the data to effectively distinguish low-quality data from high-quality data. To avoid ambiguity here, we have replaced "a certain level of noise" with "a large level of noise" in the revised version.

---

> ### Author Response · Authors · 2024-12-03
> **In Figure 1, how are the rates computed?**
>
> >*In Figure 1, how are the rates computed? and why do they show a higher sensitivity of data valuation methods to outliers?*
>
> Re:
> 1. **How to compute normalized outlier detection rate and in-distribution data detection rate?**
>
> To evaluate outlier detection, we rank data points by their value scores in ascending order and select a subset with the lowest scores. The selection precision, defined as the percentage of corrupted points in the subset, represents the outlier detection rate. Similarly, for in-distribution data detection, we rank data points by their value scores in descending order and select a subset with the highest scores. The selection precision, defined as the percentage of uncorrupted points, represents the in-distribution detection rate.
>
> To account for potential imbalance between corrupted and uncorrupted data, we normalize both detection rates. First, we compute the improvement over the precision of random selection baseline. Next, we divide this improvement by the maximum achievable gain, defined as the difference in precision between optimal selection and the random selection baseline. For illustration, consider a training set of 10,000 data points, with 50% corrupted and 50% uncorrupted. For a given selection budget, improvements in in-distribution data and outlier detection are measured against the 50% precision of random selection baseline (i.e., the corrupted and uncorrupted ratios). If the selection budget is 5,000 or less, the optimal selection precision is set to 1.0. If the selection budget exceeds 5,000, the optimal selection precision is computed as the ratio of 5,000 to the selection budget.
>
> Note that we use 10,000 data points as the training set and 1,000 data points as the validation set, randomly sampled from the training and test data of CIFAR-10, respectively. For LOO method, the training set is reduced to 5,000 and the validation set to 500, to save runtime. To visualize the difference between the normalized outlier detection rate (NDR_outlier) and the normalized in-distribution data detection rate (NDR_in-distribution), we inspect the top 20% of data points, except for influence function based method, where the top 40% is considered for a complete display.
>
> 2. **Why do they show a higher sensitivity of data valuation methods to outliers?**
>
> Specifically, for optimal transport based data valuation method, this phenomenon can be attributed to the inherent property of the Wasserstein distance of its highly unrobust to outliers [1, 2]. We will elaborate more on the reason in our theoretical analysis part.
>
> >[1] Mukherjee, D., Guha, A., Solomon, J. M., Sun, Y., & Yurochkin, M. (2021, July). Outlier-robust optimal transport. In International Conference on Machine Learning (pp. 7850-7860). PMLR. \
> >[2] Fatras, K., Séjourné, T., Flamary, R., & Courty, N. (2021, July). Unbalanced minibatch optimal transport; applications to domain adaptation. In International Conference on Machine Learning (pp. 3186-3197). PMLR.

---

> ### Author Response · Authors · 2024-12-03
> **The principled and general way to perform feature extraction**
>
> >*Instead of ResNet-18 and ResNet-18 pretrained on Image1K, what do you propose as a principled and general way to perform feature extraction for a dataset to use your method?*
>
> Re: We adopt a general approach to feature extraction that aligns with practices in existing literature (e.g., [1, 2, 3]). Specifically, in the first round of data valuation, using a simple neural network trained on the validation set is consistent with previous studies. However, in the second round of data valuation, it is important to address the risk of prior knowledge from the clean validation set which inadvertently influences the alignment of the new noisy validation set with the remaining training samples. Given our limited access to noisy data, leveraging a general pretraining neural network to extract features proves sufficient. For image datasets, the most extensively studied data type in our scenarios, under this principle, we simply take a ResNet-18 and a ResNet-18 pretrained on ImageNet1K, both of which are widely adopted for processing image datasets.
>
> For tabular data, we have added related experiments on the paper. Still consistent with prior literature (e.g., [4, 5]), we do not need the feature extraction in our framework for two rounds of data valuation. For other more complex datasets, exploring advanced feature extraction methods is left as a direction for future work, related to the field of representation learning.
>
> >[1] Sorscher, B., Geirhos, R., Shekhar, S., Ganguli, S., & Morcos, A. (2022). Beyond neural scaling laws: beating power law scaling via data pruning. Advances in Neural Information Processing Systems, 35, 19523-19536. \
> >[2] Just, H. A., Kang, F., Wang, J. T., Zeng, Y., Ko, M., Jin, M., & Jia, R. (2023). Lava: Data valuation without pre-specified learning algorithms. arXiv preprint arXiv:2305.00054. \
> >[3] Xia, X., Liu, J., Yu, J., Shen, X., Han, B., & Liu, T. (2022, September). Moderate coreset: A universal method of data selection for real-world data-efficient deep learning. In The Eleventh International Conference on Learning Representations. \
> >[4] Jiang, K., Liang, W., Zou, J. Y., & Kwon, Y. (2023). Opendataval: a unified benchmark for data valuation. Advances in Neural Information Processing Systems, 36. \
> >[5] Kwon, Y., & Zou, J. (2021). Beta shapley: a unified and noise-reduced data valuation framework for machine learning. arXiv preprint arXiv:2110.14049.

---

> ### Author Response · Authors · 2024-12-04
> **Thanks for your reply**
>
> Thanks for your reply!
>
> In a general data valuation framework, a clean and high-quality validation set is typically provided and the goal is to assess the contribution of each individual training data points to the overall performance on this provided validation set. Therefore, data valuation formalizes data selection by identifying a subset of training set to include or exclude based on alignment with the target distribution represented by the validation set, incorporating validation performance into individual training samples. We have considered such the very general data valuation framework as defined in previous papers [1] and we leverage the key effect of validation set here to augment existing data valuation methods. Data-oob (the out-of-bag estimate data valuation method) is the only one very special exception that falls out the scope of the general data valuation framework which does not utilize the validation set. Therefore, we cannot adapt Data-oob to our framework. For all other data valuation methods mostly used with satisfying results and efficiency, we have included in our paper.
>
> We are sorry that at that time, we are still waiting for some experiment results. I hope now we all specifically address your concerns regarding our paper.
>
> With best regards,
>
> Authors of Paper10029
>
> >[1] Jiang, K., Liang, W., Zou, J. Y., & Kwon, Y. (2023). Opendataval: a unified benchmark for data valuation. Advances in Neural Information Processing Systems, 36.

---

### Official Review · Reviewer_tNfM · 2024-11-03

**Soundness:** 3
**Presentation:** 3
**Contribution:** 3
**Rating:** 8
**Confidence:** 3

**Summary:**

The authors propose an interesting and novel way of computing data values, specifically aimed at ensuring high sensitivity to high-quality data and low-quality data (outliers), crucial for the data selection use case.

**Strengths:**

- The authors clearly state the problem, and Figure 1 helps show the noise sensitivity levels for the different methods.
- The suggested solution is novel and well-illustrated.
- It was good to see authors assess JST with semi-value approaches like data Banzhaf and show where JST may not do so well.
- The figures (both in the results and methodology section) that the authors use are very descriptive, nicely drawn, and intuitive.
- The authors run experiments on various image datasets: MNIST, CIFAR, SVHN, and Food-101.

**Weaknesses:**

**Questions and weaknesses**
- The authors could improve the figures and captioning, especially in the main text. The captions are not sufficiently descriptive. For example, in Figure 3, the authors don't clearly state the base data valuation method used.  In Figure 6, when using the random data valuation, it's unclear what the selection criteria for the images is. And what's the base data valuation method used?
- Could authors provide the rationale for selecting varied noise levels for different experiments (different base methods) on the same dataset? For example, why is noise level 10 in Figure 8 but 6/9 in Figure 7, and so on?
- I wonder if the initial utility function that uses the “general clean validation” set might in principle be “significantly” different from the one that uses the “low-quality-noisy validation” set. Consequently, I am curious if and by how much the value of a datum changes in the initial setup versus the second setup. If the value of datum A > the value of datum B in the initial setup is that still the case?
- What’s the stability of JST across runs?
- In case of issues like replication, wouldn’t that “poison” the level-two training/validation sets?
- Without “synthetic” noise injection, how well does JST work or compare to marginal-contribution methods on “natural” data with likely outliers, for example, low representation of samples from a given class, sensitive group, etc?

**Miscellaneous and minor**
- Minor: The text font size on the figures could be made larger to improve readability.
- How well does the framework scale to tabular datasets?
- Are there differences in class balance when using JST versus the base/pure data valuation method? Is it likely to make things worse/better?
- Given the costly nature of level-one data valuation, how scalable is two-level data valuation (JST)?
- Given that JST doesn’t improve things in methods like Banzhaf values, where these methods have additional favorable characteristics like replication robustness and consistency across runs, what’s the incentive for one to use JST over them?

**Questions:**

Questions are contextually embedded in the weakness section. Authors should please address questions added in the sub-section "questions and weaknesses".

---

> ### Author Response · Authors · 2024-12-04
> **Words of Appreciation**
>
> We would like to thank You for the positive assessment! We answer the questions below and hope they can address your concerns.

---

> ### Author Response · Authors · 2024-12-04
> **Questions and weaknesses [part 1/2]**
>
> >*The authors could improve the figures and captioning, especially in the main text. The captions are not sufficiently descriptive. For example, in Figure 3, the authors don't clearly state the base data valuation method used. In Figure 6, when using the random data valuation, it's unclear what the selection criteria for the images is. And what's the base data valuation method used?*
>
> Re: Regarding the base data valuation method used in Figure 3 and Figure 6, we use optimal transport based data valuation method to instantiate our framework. For the random selection baseline criteria for the images in Figure 6, we shuffle the dataset and show the first 5 images. We have refined all captions to be more descriptive in our updated version of paper.
>
> >*Could authors provide the rationale for selecting varied noise levels for different experiments (different base methods) on the same dataset? For example, why is noise level 10 in Figure 8 but 6/9 in Figure 7, and so on?*
>
> Re: Our goal is to improve the sensitivity of existing data valuation methods to high-quality data, particularly under the challenging small noise conditions. Therefore, for CIFAR-10 and CIFAR-100, we select small noise levels of the standard deviation based on the specific range previously used by each existing data valuation method. This is the reason we use 3/6 for optimal transport based data valuation method and use 6/9 for influence function based and reinforcement learning based data valuation method.
>
> Regarding Figure 8, we intentionally use a higher noise level of 10 to identify and discuss the failure cases of our framework. Typically, larger noise levels make it easier for data valuation methods to distinguish between in-distribution data and outliers. However, even under these favorable conditions, these marginal contribution based data valuation methods augmented by our framework do not improve the pure data valuation methods largely due to their lack of the property of higher sensitivity to outliers.

---

> ### Author Response · Authors · 2024-12-04
> **Questions and weaknesses [part 2/2]**
>
> >*I wonder if the initial utility function that uses the “general clean validation” set might in principle be “significantly” different from the one that uses the “low-quality-noisy validation” set. Consequently, I am curious if and by how much the value of a datum changes in the initial setup versus the second setup. If the value of datum A > the value of datum B in the initial setup is that still the case?*
>
> Re: To compare the initial utility function in the first round of data valuation with the one in the second round of data valuation, we compute the mean of data values of corrupted low-quality data and uncorrupted high-quality data and also the mean of data values of all data points for each round. For the first round of data valuation, we also compute these statistics after removing the data points selected as the validation set for the second round as one fair comparison. As shown in Section A (https://anonymous.4open.science/r/JST-005F/JST_main_track_tNfM_reply.pdf), the utility function changes a lot because we reverse the “general clean validation” set to the “low-quality-noisy validation” set. Accordingly, the utility function also reverses.
>
> After negating the utility function in the second round of data valuation, we evaluate if the value of datum A > the value of datum B in the initial setup is that still the case in the second round. After removing the data points selected as the validation set for the second round, we compute the number of pairs of high quality data greater than low quality data in the first round and in the second round. Also, we compute the number of pairs of high quality data greater than low quality data in the first round remaining in the second round. As shown in Section A (https://anonymous.4open.science/r/JST-005F/JST_main_track_tNfM_reply.pdf), although our framework losses some pairs, it remains most desired pairs of high quality data greater than low quality data and also improves the total number of desired pairs in the second round.
>
> >*What’s the stability of JST across runs?*
>
> Re: Optimal transport based data valuation method is implemented as a deterministic algorithm, so it shows no variation in results across runs.
>
> For influence function based and reinforcement learning based methods, the stability of JST is closely tied to the stability of the data valuation method augmented within it. However, JST consistently outperforms the first round (i.e., the pure data valuation method). In Section B (https://anonymous.4open.science/r/JST-005F/JST_main_track_tNfM_reply.pdf), we have reported the mean and standard deviation across five runs on dataset CIFAR-10.
>
> >*In case of issues like replication, wouldn’t that “poison” the level-two training/validation sets?*
>
> Re: Optimal transport based, influence function based, and reinforcement learning based data valuation methods do not consider the robustness to duplication. When data points are directly copied, these methods assign identical scores to the duplicates. Consequently, in cases of replication, the duplicates with identical scores are split together into the level-two training set or validation set. After the second round of data valuation, these duplicates continue to receive identical scores due to the lack of robustness to duplication by these methods.
>
> >*Without “synthetic” noise injection, how well does JST work or compare to marginal-contribution methods on “natural” data with likely outliers, for example, low representation of samples from a given class, sensitive group, etc?*
>
> Re: We utilize the dataset CIFAR-10-Warehouse [1], which allows us to mix original CIFAR-10 dataset with other feature styles of CIFAR-10 variants. In this way, we create a long-tail distribution of mixed CIFAR-10 dataset with other feature styles of CIFAR-10 variants as the low representation of samples from a sensitive group and we consider these samples as outliers. The proportion of original CIFAR-10 dataset to other feature styles of CIFAR-10 variants is 9:1. The experimental setting maintains as the same of our paper described. We have shown the results in Section C (https://anonymous.4open.science/r/JST-005F/JST_main_track_tNfM_reply.pdf). Still, our JST framework does not improve largely compared with the pure data valuation methods for marginal contribution based data valuation methods. Due to time constraint, we push more experiments for this question after the rebuttal period.
>
> >[1] Sun, X., Leng, X., Wang, Z., Yang, Y., Huang, Z., & Zheng, L. (2023). Cifar-10-warehouse: Broad and more realistic testbeds in model generalization analysis. *arXiv preprint arXiv:2310.04414*.

---

> ### Author Response · Authors · 2024-12-04
> **Miscellaneous and minor [part 1/2]**
>
> >*Minor: The text font size on the figures could be made larger to improve readability.*
>
> Re: Due to time constraint, we will adjust all figures in the final version to increase font size for improved readability.
>
> >*How well does the framework scale to tabular datasets?*
>
> Re: To cover the complexity of datasets we evaluate on, we have included a time-series tabular dataset HHAR [1] for the task of cross-domain retrieval. Based on prior studies [2, 3], we divide HHAR sequential data into four domains based on devices and motion sensors used, using each domain as the validation set while treating the training set as a mixed-domain dataset. Based on previous studies [4, 5], we apply the k-means clustering algorithm to the data values and partition the training set into two groups. We identify the cluster with the higher average data value as the potential target domain data for retrieval. Then to evaluate our JST framework, we compare clustering results against ground truth and calculate the F1-score. We compare our JST framework with the pure data valuation method. Note that for a fair comparison, we calculate the F1-score using only the data propagated into the second round of our JST framework for the pure data valuation method.
>
> For tabular data type, feature extraction is not required, as noted in prior studies [4, 5]. For influence function based and reinforcement learning based data valuation methods, we follow previous studies [2, 3] to use a 1D-CNN model as the dependent model and to reduce the running time, we randomly sample 10% of data to utilize. The results run independently and repeatedly in 10 times and we report the ’average±standard error’. Boldface numbers denote the best method.
>
> As demonstrated in Section D (https://anonymous.4open.science/r/JST-005F/JST_main_track_tNfM_reply.pdf), our JST framework consistently outperforms the pure data valuation method, even under conditions of severe domain imbalance. Specifically, in the HHAR dataset, the balance score — calculated as the average proportion of the minority domain to the majority domain when selecting two random domains — is as low as 47\%.
>
> >*Are there differences in class balance when using JST versus the base/pure data valuation method? Is it likely to make things worse/better?*
>
> Re: Based on prior literature [6], we calculate the class balance score b ∈ [0%, 100%] as the average class imbalance across any two pairs of classes by computing the expectation over taking two random classes, and then computing how many images the minority class has in proportion to the majority class. We evaluate our approach on dataset CIFAR-10 and employ our JST framework on three methods (optimal transport based, influence function based and reinforcement learning based) included in the paper with consistent relatively large standard deviation (6, 9, and 9, respectively). As shown in Section E (https://anonymous.4open.science/r/JST-005F/JST_main_track_tNfM_reply.pdf), we maintain a similar level of balance score compared with pure data valuation methods across different numbers of data points selected.
>
> >[1] Stisen, A., Blunck, H., Bhattacharya, S., Prentow, T. S., Kjærgaard, M. B., Dey, A., ... & Jensen, M. M. (2015, November). Smart devices are different: Assessing and mitigatingmobile sensing heterogeneities for activity recognition. In *Proceedings of the 13th ACM conference on embedded networked sensor systems* (pp. 127-140). \
> >[2] Ragab, M., Eldele, E., Tan, W. L., Foo, C. S., Chen, Z., Wu, M., ... & Li, X. (2023). Adatime: A benchmarking suite for domain adaptation on time series data. *ACM Transactions on Knowledge Discovery from Data*, *17*(8), 1-18.\
> >[3] He, H., Queen, O., Koker, T., Cuevas, C., Tsiligkaridis, T., & Zitnik, M. (2023, July). Domain adaptation for time series under feature and label shifts. In *International Conference on Machine Learning* (pp. 12746-12774). PMLR.\
> >[4] Kwon, Y., & Zou, J. (2021). Beta shapley: a unified and noise-reduced data valuation framework for machine learning. arXiv preprint arXiv:2110.14049.\
> >[5] Jiang, K., Liang, W., Zou, J. Y., & Kwon, Y. (2023). Opendataval: a unified benchmark for data valuation. Advances in Neural Information Processing Systems, 36. \
> >[6] Sorscher, B., Geirhos, R., Shekhar, S., Ganguli, S., & Morcos, A. (2022). Beyond neural scaling laws: beating power law scaling via data pruning. Advances in Neural Information Processing Systems, 35, 19523-19536.

---

> ### Author Response · Authors · 2024-12-04
> **Miscellaneous and minor [part 2/2]**
>
> >*Given the costly nature of level-one data valuation, how scalable is two-level data valuation (JST)?*
>
> Re: Our method directly takes over the results from the first-stage ranking, only doubling the time complexity from O(n) to O(2n) without adding any additional overhead. Therefore, the overall time complexity is determined by the method used to instantiate our framework. For the experiments in raw dataset pruning, leveraging the exceptional efficiency and near-linear time complexity of the optimal transport based method, our JST framework achieves data valuation in a remarkably short time, completing in just a few seconds as shown in the table.
>
> | **JST on Datasets** | **Time** |
> | --- | --- |
> | SVHN - Round 1 | 9.9 sec. |
> | SVHN - Total | 21.6 sec. |
> | Food101 - Round 1 | 44.4 sec. |
> | Food101 - Total | 90.7 sec. |
>
> Additionally, we analyze the time complexity of three methods—optimal transport based, influence function based, and reinforcement learning based—across data sizes ranging from 0 to 50,000. Even with doubled time complexity, our framework completes in mere seconds for the optimal transport based method. For the reinforcement learning based method, our framework takes approximately 10 minutes for 50,000 data points and around 1 minute for smaller datasets with 10,000 data points. Only for the influence function based method applied to large datasets, our framework requires several hours to finish. All results have been added to the appendix.
>
> >*Given that JST doesn’t improve things in methods like Banzhaf values, where these methods have additional favorable characteristics like replication robustness and consistency across runs, what’s the incentive for one to use JST over them?*
>
> Re: Each data valuation method has its own merits and the choice of method often depends on the specific problem setting. Compared with DataBanzhaf and AME data valuation methods, which improve the marginal contribution based method and take several hours to finish, optimal transport based and reinforcement learning based methods have very low runtime requirement. With the constraint of computing resources to choose these two data valuation methods instead of Databanzhaf and AME, our JST framework would further improve their performance greatly. Also, influence function based method has more fundamental application settings, where our JST framework can be spread into these settings.

---

### Official Review · Reviewer_5dqT · 2024-11-05

**Soundness:** 3
**Presentation:** 4
**Contribution:** 3
**Rating:** 6
**Confidence:** 5

**Summary:**

This paper proposes a two-stage framework, JST, to assist existing data valuation methods in selecting high-quality data points. These data valuation methods are sensitive to outliers but fall short in recognizing in-distribution data points. This framework uses outliers as a validation set and again leverages the sensitivity of these methods to outliers to identify high-quality data points. The experiments show that JST significantly improves over baselines, verifying the effectiveness of the framework. However, it is not adaptable for marginal contribution-based data valuation methods due to a lack of the property of higher sensitivity to outliers.

**Strengths:**

1. The narrative of this paper is clear and easy to understand, and the illustrations are vivid and illustrative.

2. The framework is experimentally effective on tested tasks.

**Weaknesses:**

1. The lack of theoretical analysis makes the article seem incomplete.

2. The results of Figure 4 seem inconsistent with the analysis of Figure 1.

3. Some experimental phenomena can be further explained.

4. I recommend that the authors add experiments about time cost to demonstrate their applicability in reality.

5. Some symbols are not clear and a bit confusing.

**Questions:**

The authors discovered a phenomenon that existing data valuation methods are not sensitive to in-distribution data points and used this property to design a two-stage framework. However, they did not explain this phenomenon mathematically.

In round 1 of Figure 4, I see that outliers are mixed with in-distribution data points at low data values. Wouldn’t this cause the NDRoutlier to be poor as well? How does JST work in this case?

In Figure 5, the three curves are very close. I think "our framework achieves much higher accuracy compared to the pure data valuation method" is overclaimed. Also, I hope the author can explain why random selection has such good results. In Figure 3, the curve of random is a horizontal line.


Adding a stage seems to more than double the time, but the improvement of the new framework in Figure 5 is not significant, which may not be a good thing in real applications. Therefore, the author can add experiments on time cost trade-offs to show that the improvement in performance is reasonable for the increase in time cost.

 “-” in the subtitles on the left and right sides of the three figures in Figure 1 is very close to “Noise Level - Std. Dev” and “NDRoutlier-NDRin-distribution”. This is a bit confusing for me to understand the whole figure.

---

> ### Author Response · Authors · 2024-11-25
> **The results of Figure 4 seem inconsistent with the analysis of Figure 1.**
>
> > *In round 1 of Figure 4, I see that outliers are mixed with in-distribution data points at low data values. Wouldn’t this cause the NDRoutlier to be poor as well? How does JST work in this case?*
>
> Re: We apologize for the lack of clarification here. In the optimal transport based data valuation method, the “data value” of a data point is defined as the derivative of the optimal transport cost with respect to the addition of that data point. A lower data value indicates that the data point reduces the optimal transport cost, which, counterintuitively, corresponds to a **higher valuation** for that data point. To ensure consistency throughout the paper, we have negated all data values (i.e., multiplied them by -1) for optimal transport based data valuation method, so a **higher data value** now directly reflects a higher valuation. We have updated the figure in the paper to reflect this adjustment. In this way, at low data values of round 1, the set predominantly comprises noisy data points, albeit with some in-distribution data points mixed in, aligning with our framework.

---

> ### Author Response · Authors · 2024-11-25
> **Some experimental phenomena can be further explained.**
>
> > *In Figure 5, the three curves are very close. I think "our framework achieves much higher accuracy compared to the pure data valuation method" is overclaimed. Also, I hope the author can explain why random selection has such good results. In Figure 3, the curve of random is a horizontal line.*
>
> Re:
> 1. **On the Claim of Higher Accuracy Being Overstated:**
>
>     First, the figure in our paper presents results across a wide range of pruning ratios, from very low to very high, leading to the y-axis covering a broad range of top-1 accuracy values, from very high to very low. In particular, for the Food-101 dataset, the second and third plots span accuracy ranges of 0–80% and 30–80%, respectively. This broad scale may make the improvements appear visually close to the random baseline, despite their significance. To address this, we will zoom in on the relevant accuracy ranges and update the figure to more clearly highlight the improvements achieved by our method.
>
>     Second, in the context of data pruning, achieving higher accuracy with less data is a challenging task, and even small improvements in accuracy are considered significant, as highlighted in previous studies [1, 2]. Similar to Figure 2 in [1] and Figures 4B, 5B, and 5C in [2], where methods are compared against random baseline and other pruning techniques, our method maintains the same level improvement as these methods for several accuracy upgrades. These improvements are substantial within the field and justify our claim.
>
> 2. **Explanation for Good Results of Random Selection:**
>
>     Random selection often yields strong results in data pruning tasks because it preserves the original data distribution and maintains class balance without introducing bias. In contrast, some specific data pruning methods may inadvertently focus on particular subsets of data, leading to class imbalance and reduced model performance. As highlighted in previous studies, including Figure 2 in [1] and Figures 4B, 5B, and 5C in [2], random pruning outperforms many other pruning techniques, emphasizing the challenge of data pruning tasks. By visualizing the top and bottom images for the random baseline, pure data valuation method and our method, we show that the pure data valuation method tends to select images of the same pattern, which introduces bias and harms the performance. However, our method selects diverse, relevant images and excludes low-quality, out-of-distribution samples, thereby outperforming both pure data valuation method and random baseline.
>
> 3. **Regarding the Random Curve as a Horizontal Line in Figure 3:**
>
>     In our scenario of selecting high-quality data, defining the random baseline as the selection precision at top-k is a standard and reasonable approach. Additionally, representing the random baseline as an expected value (i.e., a horizontal line), rather than simulating random selection, is a common practice in the literature [3, 4]. This approach provides a clear reference point, simplifying comparisons and enabling easier evaluation of our method’s effectiveness relative to the baseline.
>
> > [1] Xia, X., Liu, J., Yu, J., Shen, X., Han, B., & Liu, T. (2022, September). Moderate coreset: A universal method of data selection for real-world data-efficient deep learning. In *The Eleventh International Conference on Learning Representations*.
> > [2] Sorscher, B., Geirhos, R., Shekhar, S., Ganguli, S., & Morcos, A. (2022). Beyond neural scaling laws: beating power law scaling via data pruning. *Advances in Neural Information Processing Systems*, *35*, 19523-19536.
> > [3] Yoon, J., Arik, S., & Pfister, T. (2020, November). Data valuation using reinforcement learning. In *International Conference on Machine Learning* (pp. 10842-10851). PMLR.
> > [4] Just, H. A., Kang, F., Wang, J. T., Zeng, Y., Ko, M., Jin, M., & Jia, R. (2023). Lava: Data valuation without pre-specified learning algorithms. *arXiv preprint arXiv:2305.00054*.

---

> > ### Author Response · Authors · 2024-11-30
> >
> > We have included a figure in Appendix D to clearly illustrate the extent of our improvements. We plot the differences on accuracy between our framework and both the pure data valuation method and the random selection baseline, highlighting the significant improvements achieved by JST framework.

---

> ### Author Response · Authors · 2024-11-25
> **I recommend that the authors add experiments about time cost to demonstrate their applicability in reality.**
>
> > *Adding a stage seems to more than double the time, but the improvement of the new framework in Figure 5 is not significant, which may not be a good thing in real applications. Therefore, the author can add experiments on time cost trade-offs to show that the improvement in performance is reasonable for the increase in time cost.*
>
> Re:
> First, our improvement is significant, as discussed in the previous questions. Second, our method directly takes over the results from the first-stage ranking, only doubling the time complexity from O(n) to O(2n) without adding any additional overhead. Therefore, the overall time complexity is determined by the method used to instantiate our framework. For the experiments in Figure 5, leveraging the exceptional efficiency and near-linear time complexity of the optimal transport based method [1], our JST framework achieves data valuation in a remarkably short time, completing in just a few seconds as shown in the table.
> | **JST on Datasets** | **Time** |
> | --- | --- |
> | SVHN - Round 1 | 9.9 sec. |
> | SVHN - Total | 21.6 sec. |
> | Food101 - Round 1 | 44.4 sec. |
> | Food101 - Total | 90.7 sec. |
>
> Additionally, we analyze the time complexity of three methods—optimal transport based, influence function based, and reinforcement learning based—across data sizes ranging from 0 to 50,000. Even with doubled time complexity, our framework completes in mere seconds for the optimal transport based method. For the reinforcement learning based method, our framework takes approximately 10 minutes for 50,000 data points and around 1 minute for smaller datasets with 10,000 data points. Only for the influence function based method applied to large datasets, our framework requires several hours to finish. All results have been added to the appendix.
>
> > [1] Just, H. A., Kang, F., Wang, J. T., Zeng, Y., Ko, M., Jin, M., & Jia, R. (2023). Lava: Data valuation without pre-specified learning algorithms. *arXiv preprint arXiv:2305.00054*.

---

> ### Author Response · Authors · 2024-11-25
> **Some symbols are not clear and a bit confusing.**
>
> >*“-” in the subtitles on the left and right sides of the three figures in Figure 1 is very close to “Noise Level - Std. Dev” and “NDRoutlier-NDRin-distribution”. This is a bit confusing for me to understand the whole figure.*
>
> Re: We apologize for the confusion here. To clarify, we have replaced the “-” in the subtitles on the left with parentheses to differentiate it from the minus symbol on the right, which represents the difference between NDRoutlier and NDRin-distribution. Figures 1 and 8B have been updated accordingly in the revised version of the paper.

---

### Author Response · Authors · 2024-12-04
**A letter to the AC and all reviewers**

We sincerely appreciate the hard work of dedicated reviewers for helping review our manuscript and providing helpful comments and valuable feedback. We are deeply grateful for your support throughout this process.

To support future advancements in the field, we have open-sourced our evaluation framework. A temporary code repository is available here: https://anonymous.4open.science/r/JST-005F/JST/README.md. Additionally, to strengthen the theoretical foundation of our work, we have included a theoretical analysis, accessible here: https://anonymous.4open.science/r/JST-005F/JST_theoretical_analysis.pdf.

We look forward to the exploration and discovery of further advancements in this line of research.

With best regards,

Authors of Paper10029

---

### Meta-Review · Area_Chair_J7A9 · 2024-12-20

**Metareview:**

This paper introduces a new procedure for data-valuation.

Given a valuation function that determines high vs low-quality points of a data-set, the two step procedure goes as follows.
Select the low-quality data-points in a first step, then value the remaining data-points of the data-set wrt the low-quality points selected. The final value is minus the latter.

The underlying assumption is that, I guess, outliers of outliers are inliers.

That sort of makes sense in some cases, but I would like to be convinced that this is appropriate under some structural assumptions. This is the main criticisms of the reviewers: can you convince us that this procedure makes sense with some theoretical guarantees ? The figures indicate that it should be the case, but why ?

Any theoretical guarantees would be a very strong addition to the paper for the future versions of the paper.

**Additional Comments On Reviewer Discussion:**

There was a small discussion about the (lack of) theoretical contribution/guarantees of this paper, which lead to my decision.

---

### Decision · Program_Chairs · 2025-01-22

Reject